# Efficient Interactive Maximization of BP and Weakly Submodular Objectives

**Adhyyan Narang**[1]  **Omid Sadeghi**[2]  **Lillian Ratliff**[1]  **Maryam Fazel**[1]  **Jeff Bilmes**[1]

[1]Electrical and Computer Engineering, University of Washington, Seattle, Washington, USA
[2] Sloan School of Management, Massachusetts Institute of Technology, Boston, MA, USA

## Abstract

In the context of online interactive machine learning with combinatorial objectives, we extend purely submodular prior work to more general non-submodular objectives. This includes: (1) those that are additively decomposable into a sum of two terms (a monotone submodular and monotone supermodular term, known as a BP decomposition); and (2) those that are only weakly submodular. In both cases, this allows representing not only competitive (submodular) but also complementary (supermodular) relationships between objects, enhancing this setting to a broader range of applications (e.g., movie recommendations, medical treatments, etc.) where this is beneficial. In the two-term case, moreover, we study not only the more typical monolithic feedback approach but also a novel framework where feedback is available separately for each term. With real-world practicality and scalability in mind, we integrate Nyström sketching techniques to significantly reduce the computational cost, including for the purely submodular case. In the Gaussian process contextual bandits setting, we show sub-linear theoretical regret bounds in all cases. We also empirically show good applicability to recommendation systems and data subset selection. The code for this paper is available at: https://github.com/AdhyyanNarang/online_bp.

## 1 INTRODUCTION

Many machine learning paradigms are offline, where a learner must understand the associations and relationships within a dataset that is gathered, fixed, and then presented. Interactive learning, by contrast, involves a dynamic, repeated, and potentially everlasting interaction between the algorithm (learner) and the environment (teacher), better mimicking natural organisms as they proceed through life. Interactive learning is quite important for applications such as recommender systems [Mary et al., 2015], natural language and speech processing [Ouyang et al., 2022], interactive computer vision [Le et al., 2022], advertisement placement [Schwartz et al., 2017], environmental monitoring [Srivastava et al., 2014], personalized medicine [Durand et al., 2018], adaptive website optimization [White, 2013], and robotics [Kober et al., 2013], to name only a few.

The fundamental mathematical challenge in these settings is to optimize a utility function that encapsulates the value or payoff of different actions within a specific context. While there are many instances within this paradigm, including reinforcement, active, online, and human-in-the-Loop (HitL) learning, one such setting is contextual bandits. In contextual bandits, the agent observes a set of features (a context vector), takes an action and then gets a reward from the environment. The goal is to maximize the total accumulated reward over a series of actions over time. The context significantly influences the optimal choice of action. For example, in movie recommendation systems, the context might include user demographics (even a specific user), past viewing history, time of day, and so on.

Gaussian Process Contextual Bandits (GPCB) extend this basic idea by incorporating Gaussian Processes (GPs) for modeling the unknown reward function [Seeger et al., 2008, Srinivas et al., 2010, Krause and Ong, 2011, Valko et al., 2013, Camilleri et al., 2021]. This approach is particularly effective in scenarios where the relationship between the context, actions, and rewards is complex and non-linear. For any given context $\phi_{u_t}$ (where $u_t \in [m]$ is the index of one of $m \in \mathbb{Z}_+ \cup \{\infty\}$ possible contexts at time $t$), GPs allow the easy expression of a posterior distribution based on previous rounds in terms of Gaussian conditional mean $\mu_{\phi_{u_t}}$ and condition variance $\sigma^2_{\phi_{u_t}}$ vectors where the former encodes value and the latter encodes uncertainty. These are combined in Upper Confidence Bound (UCB) algorithms as $\mu_{\phi_{u_t}}(v) + \beta_t \sigma^2_{\phi_{u_t}}(v)$ to offer a combined valuation of

| | Offline | Pure Online | Online + Nyström | Online + Sep. FB | Online + Nyström + Sep. FB |
|---|---|---|---|---|---|
| **Modular** | | Srinivas et al. [2010], Krause and Ong [2011] | Zenati et al. [2022] | N/A | N/A |
| **SM** | Nemhauser et al. [1978] | Chen et al. [2017] | ✓ | N/A | N/A |
| **BP** | Bai and Bilmes [2018] | ✓ | ✓ | ✓ | ✓ |
| **WS** | Das and Kempe [2011], Bian et al. [2019] | ✓ | ✓ | N/A | N/A |

Table 1: The present paper's novelty ( green ✓ which means new algorithms for sublinear regret) in the context of previous work. Here **SM** refers to SubModular, **BP** to suBmodular-suPermodular, and **WS** to Weakly Submodular. **Sep FB** refers to the separate feedback BP setting introduced in this paper. N/A means not applicable.

action $v \in V$ in the context of $\phi_{u_t}$ in terms of exploration (high $\sigma^2_{\phi_{u_t}}(v)$) vs. exploitation (high $\mu_{\phi_{u_t}}(v)$) where $\beta_t$ is a computed time-dependent tradeoff coefficient. The goal of GPCB is traditionally to minimize cumulative regret, where the rewards at each time are compared to the best choice at that time:

$$\mathcal{R}(T) = \sum_{t=1}^{T} f_{\phi_{u_t}}(v^*_{\phi_{u_t}}) - f_{\phi_{u_t}}(v_t),$$

where $v^*_{\phi_{u_t}}$ is the best choice for context $\phi_{u_t}$ and $v_t$ is the algorithm's choice at time $t$. Sublinear regret means this increases more slowly than $T$ (i.e., $\lim_{T\to\infty} \mathcal{R}(T)/T = 0$).

Chen et al. [2017] made the important observation that GPCBs can be used for online combinatorial, specifically submodular, maximization, where an input set of size $T$ is incrementally constructed over time. Offline monotone submodular maximization is NP-hard but a greedy algorithm has an $\alpha$ multiplicative approximation [Nemhauser et al., 1978] for $\alpha = 1 - 1/e$. Chen et al. [2017] utilizes $\alpha$-regret, where the regret of the online algorithm at time $T$ is based on comparing with the $\alpha$ approximation of the offline algorithm, specifically

$$\mathcal{R}(T) = \alpha \sum_{q=1}^{m} f_{\phi_q}(S^*_q) - f_{\phi_q}(S_{T_q,q}),$$

where $f_{\phi_q}$ is the submodular function for context $\phi_q$ having $S^*_q$ as the optimal solution and $S_{T_q,q}$ is the algorithm's incrementally-computed attempted solution both of size $|S^*_q| = |S_{T_q,q}| = T_q$, and $T = \sum_q T_q$ where $T_q$ is the frequency of context $q$. Thus, unlike the standard GPCP above which uses a summation of pointwise quantities, this combinatorial $\alpha$-regret utilizes the interdependencies between elements evaluated by the submodular function. These interdependencies strongly influence the best choices at different time steps because of the submodular (i.e., non-independent) relationships. Another critical feature is that the function $f$ is not available to the algorithm — rather only noisy gain queries $y_t$ of the form $y_t = f(v|S_t) + \epsilon_t$ are available *after* $v$ has been committed, where $v$ is the algorithmic choice, $S_t = \{v_1, v_2, \ldots, v_{t-1}\}$ constitutes the previous and now fixed set of choices, and $\epsilon_t$ is independent noise. Compared to the offline setting, the online optimization setting becomes significantly more mathematically challenging and requires smoothness assumptions to achieve sublinear regret. However, the online setting is natural for many applications.

Despite its many benefits, the purely submodular assump-

tion is not sufficiently expressive to capture essential properties of many real-world environments. Consider the example of movie recommendations — in some cases, recommending a movie and its sequel will yield greater rewards than recommending the two movies independently, a complementarity (i.e., supermodularity) amongst actions. In personalized medicine, certain combinations of medicines might together possess pharmacological synergy (a supermodularity) while other combinations will be lethal (a submodularity).

**Contributions.**   In the present paper, we offer results that achieve sublinear $\alpha$-regret in the GPCB setting for a variety of non-linear non-submodular utility functions that previously have not been considered in the online setting.

We first consider when non-submodular utility functions $h = f + g$ can be additively decomposed into the sum of a monotone submodular $f$ and monotone supermodular $g$ components, known as a "BP" function (Definition 2). BP functions allow for a much more expressive representation of utility, capturing both the diminishing returns inherent in submodular and the increasing complementary returns characteristic of supermodular functions. Bai and Bilmes [2018] introduced and studied the offline maximization of BP functions subject to a cardinality constraint — this was shown to have an approximation ratio of $\alpha = \frac{1}{\kappa_f}\left[1 - e^{-(1-\kappa^g)\kappa_f}\right]$ where $\kappa_f, \kappa^g \in [0, 1]$ are the submodular and supermodular curvatures that respectively measure how far the functions $f, g$ are from being modular (see Section B). In general, this problem is inapproximable, but if $\kappa^g < 1$, it is possible to obtain approximation ratios for this problem using the greedy algorithm. We study this problem in the $\alpha$ regret case showing sublinear regret. More interestingly, this decomposition enables us to study a novel form of *separate feedback* where we receive separate rewards each for the submodular $f$ and supermodular $g$ components. In an interactive recommender system, for example, the utility function might represent the combined effects of user satisfaction (submodular due to saturation of interests) and network effects (supermodular due to the increasing value of shared community experiences) the rewards each of which can be available separately. The stronger separate feedback case allows us to provide a stronger $\alpha$-regret with $\alpha = \min\left\{1 - \frac{\kappa_f}{e}, 1 - \kappa^g\right\}$. This choice is inspired by Liu et al. [2022], who proposed a distorted version of the offline greedy algorithm for BP maximization problems and provided an improved $\min\{1 - \frac{\kappa_f}{e}, 1 - \kappa^g\}$ approximation ratio.

See Appendix H.1 for further commentary on this approach.

When $h : 2^V \to \mathbb{R}$ is not decomposable as with a BP function, we next consider a monolithic $h$ that is *Weakly Submodular* (WS), defined as the following being true: $\forall A \subseteq B \subseteq V, \sum_{b \in B \setminus A} h(b|A) \geq \lambda h(B|A)$ for some $\lambda \in [0, 1]$ where $h(B|A) \triangleq h(B \cup A) - h(A)$. The largest $\lambda$ for which $h$ is weakly submodular is known as the submodularity ratio $\gamma$ [Das and Kempe, 2011, Calandriello et al., 2018], and $h$ is submodular if and only if $h$ is 1-weakly submodular. Bian et al. [2019] also introduced a generalized version of the submodular curvature $\zeta$ for WS functions and studied the approximation ratio of the offline greedy algorithm on such functions. Inspired by these results, we present a sublinear regret bound on WS functions with $\alpha = \frac{1}{\zeta}\big(1 - e^{-\zeta\gamma}\big)$.

We remark that in general, just because an offline algorithm can achieve an $\alpha$-approximate solution to an NP-hard problem does not guarantee that the online GPCB version can achieve sublinear $\alpha$-regret — it is in general quite challenging to show sublinear $\alpha$-regret for new problems especially in the combinatorial case when there are such dependencies between previous and current actions.

A third contribution of our paper further addresses the main practical computational complexity challenge with GPs, especially in high-dimensional spaces. The problem arises from performing operations on Gram covariance matrices, whose shape increase as the number of observations grow. We address this challenge, for both the $h$ as a BP function and $h$ as a WS function cases, by showing the applicability of Nyström approximations, a technique traditionally used in kernelized learning to efficiently handle large-scale data. We show that Nyström approximations facilitate the efficient computation of our utility functions by approximating its components in a lower-dimensional space. This method significantly reduces the computational complexity from the prohibitive $O(T^3)$ to a more manageable form, typically $O(CTN^2)$ where $N$ is substantially smaller than $T$ and represents the number of points in the Nyström approximation.

Lastly, in our numerical experiments, we empirically demonstrate the above for two applications, movie recommendations and in machine learning training data subset selection.

**Background and Other Related Work.** The above introduces this paper's novel contributions in the context of previous work which Table 1 briefly summarizes. A very detailed literature review is given in Appendix C. We highlight that our algorithms are inspired by the developments in GPCBs that enable the optimization of unknown functions in low-information online environments [Srinivas et al., 2010, Krause and Ong, 2011, Valko et al., 2013, Camilleri et al., 2021]. In particular, Zenati et al. [2022] improves computational efficiency by using Nyström points to speed up the algorithm with the same asymptotic regret guarantee as

prior work. Additionally, there is a line of work on "combinatorial bandits" that may appear similar to our formulation at first glance [Takemori et al., 2020, Papadigenopoulos and Caramanis, 2021, Kveton et al., 2014, Chen et al., 2018, Nie et al., 2023, Streeter and Golovin, 2008] – these study the computational complexity of learning an unknown submodular function. However, the feedback model in those papers is entirely different than us: a new submodular function arrives at each time step and an entire set is recommended. In the present work, as mentioned above, we accumulate a selected set over time for functions that arrive repeatedly. Hence, this body of work is not comparable to the present work.

In the following section, we begin directly with our problem formulation. For further background on submodularity, supermodularity, BP functions, and various curvatures, see Appendix B.

## 2 PROBLEM FORMULATION

Our optimizer operates in an environment that occurs over $T$ time steps. Specifically, at each time step $t \in [T]$:

1. The optimizer encounters one of $m$ set functions from the set $\{h_1 \ldots h_m\}$ each defined over the finite ground set $V$. The optimizer is ignorant of the function but knows its index $u_t \in [m]$ as well as a context or feature-vector $\phi_{u_t}$ for that index at round $t$.

2. The optimizer computes and then performs/plays action $v_t \in V$, and then adds $v_t$ to its growing context-dependent set $S_{t_{u_t}, u_t}$ of size $|S_{t_{u_t}, u_t}| = t_{u_t}$ with $\sum_{j \in [m]} t_j = t$. The set $S_{t_{u_t}, u_t}$ contains all items so far selected for the unknown function $h_{u_t}$.

3. The environment offers the optimizer noisy marginal gain feedback. There are two feedback models:

   (3a) *Monolithic Feedback:* The optimizer receives $y_t$ with $y_t = h_{u_t}(v_t|S_{t_{u_t}, u_t}) + \epsilon_t$.

   (3b) *Separate Feedback:* In the BP case, $(y_{f,t}, y_{g,t})$ may be available with $y_{f,t} = f_{u_t}(v_t|S_{t_{u_t}, u_t}) + \epsilon_t/2$ and $y_{g,t} = g_{u_t}(v_t|S_{t_{u_t}, u_t}) + \epsilon_t/2$.

The separate feedback case (3b) is relevant only for applications (e.g., multiple surveys, etc.) where it is feasible. Section 5 exploits this richer feedback to improve performance. All feature-vectors $\phi_{u_t}$ are chosen from set $\Phi$ of size $|\Phi| = m$, and we assume that the identity of the utility function $h_q$ is determined uniquely by $\phi_q$; hence, when clear from context, we use $h_q$ to refer to $h_{\phi_q}$.

We observe how two applications may be formalized in our framework. Vignette 2 is further explored in Appendix I.

**Vignette 1** (Movie Recommendations). Each function $h_q$ captures the preferences of a single user $q \in [m]$, and the index $u_t \in [m]$ reveals which user has arrived at time step $t$.

The action $v_t$ performed at time $t$ is the optimizer's recommended movie to user $u_t$. The feature vector $\phi_{u_t}$ contains user-specific information, e.g., age range, favorite movies and genres, etc. The feedback gain $h_{u_t}(v|A)$ is the enjoyment user $u_t$ has from watching movie $v$ having already watched the movies in set $A$.

**Vignette 2** (Active Learning). The optimizer chooses training points to be labeled for $m$ related tasks on the same dataset - for instance classification, object detection, and captioning. The function $h_q(A)$ is the test accuracy of a predictor trained on set $A$ on the $q^{th}$ task. Choosing an action $v_t$ is tantamount to choosing a training point to be labeled for task $u_t \in [m]$.

In our quest to design low regret online item-selection strategies for these problems (made precise in Section 4), we first study the robustness of the greedy procedure for the offline optimization of Monotone Non-decreasing Normalized (MNN) functions (see Appendix B) in Section 3. Then in Section 4, we show that our proposed online procedure approximates the offline greedy algorithm, leveraging Section 3 to obtain online guarantees.

# 3   OFFLINE ALGORITHM ROBUSTNESS

We consider the problem of cardinality-constrained optimization of a MNN objective $h : 2^V \to \mathbb{R}$:
$$\max_{S \in 2^V} h(S) : |S| \leq k \tag{1}$$
Let $S^*$ denote an achieving set solving Equation (1). The most common approximation algorithm for this problem greedily [Nemhauser et al., 1978] maximizes the available marginal gain having oracle access to $h$. In online settings, however, we do not have this luxury. To help us analyze the online setting, therefore, we consider a modified offline algorithm where the greedy choices might be good only with respect to a set of additive "slack" variables $r_j$, exploring the impact of this modification on approximation quality for different classes of functions. Then in Section 4 we develop an online algorithm that emulates greedy in this way.

## 3.1   GREEDY SELECTION ROBUSTNESS

We define an **approximate greedy** selection rule that, given scalars $\{r_j\}_{j=1}^k$, chooses $v_j$ for each $j \in [k]$ satisfying:
$$v_j \in \{v : h(v|S_{j-1}) \geq \mathrm{argmax}_{\tilde{v}} \, h(\tilde{v}|S_{j-1}) - r_j\}, \tag{2}$$
where $S_j = \{v_1 \dots v_j\}$ and $S^*$ the optimal set of size $k$.

**Lemma 1.** *Any output $S$ of the approximate greedy selection rule in Equation (2) admits the following guarantee for BP objectives (Def. 2) for Problem (1):*
$$h(S) \geq \frac{1}{\kappa_f}\left[1 - e^{-(1-\kappa^g)\kappa_f}\right] h(S^*) - \sum_{j=1}^k r_j,$$
*where $\kappa_f, \kappa^g$ are as defined in Definitions 3 and 4.*

A proof is in Appendix F.1. This result is a generalization of Bai and Bilmes [2018, Theorem 3.7] which is recovered by setting $\forall j, r_j = 0$. This result is surprising because, with the supermodular part of the BP function, poor early selections may preclude the ability to exploit potential increasing returns from $g$ — the curvature $\kappa^g$ is crucial for this. The result can also be understood as a generalization of Chen et al. [2017], which studies the robustness of the greedy algorithm to errors in submodular functions. In their case, however, they adapt the simple classical greedy algorithm proof [Nemhauser et al., 1978]. In Appendix E, we provide an alternate proof using a crude bound that incorporates the supermodular curvature but ignores the submodular curvature, reminiscent of the argument in Chen et al. [2017]. However, the approximation ratio obtained is much worse than that of Bai and Bilmes [2018].

Therefore, we study (in Appendix F.1) the robustness using the detailed analysis in Bai and Bilmes [2018]. This poses a considerable challenge, since Bai and Bilmes [2018] (inspired by Conforti and Cornu'ejols [1984]) formulate an intricately designed series of linear programs to show that any selection that has as much overlap with the optimal solution as the greedy algorithm must achieve the desired approximation ratio. Here, the errors $r_j$ manifest as perturbations to the constraints of the linear programs. We then perform a *sensitivity analysis* of the linear programs to argue that these perturbations to the constraints lead to a linear perturbation to the optimal objective and does not cause it to explode.

In the case where $h$ does not have a BP decomposition, we offer the following result generalizing Bian et al. [2019].

**Lemma 2.** *Any output $S$ of the approximate greedy selection rule in Equation (2) admits the following guarantee on objectives with submodularity ratio $\gamma$ and generalized curvature $\zeta$ (Definitions 5 and 6) for Problem (1):*
$$h(S) \geq \frac{1}{\zeta}\left(1 - e^{-\zeta\gamma}\right)h(S^*) - \sum_{j=1}^k r_j.$$

A proof is given in Appendix F.2. We see in Section 4 that Lemmas 1 and 2 are key to the analysis of Algorithm 1.

## 3.2   DISTORTED BP GREEDY ROBUSTNESS

In Liu et al. [2022], the authors present a "distorted" version of the greedy algorithm, which achieves a better greedy approximation ratio than Bai and Bilmes [2018] for Problem (1) with a BP objective. Here, we study its robustness.

As in Sviridenko et al. [2017], we define the modular lower bound of the submodular function $l_1(S) = \sum_{j \in S} f(j|V\backslash\{j\})$. Also, define the totally normalized submodular function as $f_1(S) = f(S) - l_1(S)$. Note that $f_1$ always has curvature $\kappa_f = 1$ and also that $h(S) =$

$f_1(S) + g(S) + l_1(S)$. We define the function $\pi_j(v|A)$ as:

$$\pi_j(v|A) = \left(1 - \frac{1}{k}\right)^{k-j-1} f_1(v|A) + g(v|A) + l_1(v) \quad (3)$$

In Liu et al. [2022], the optimizer greedily maximizes the $\pi_j$ function at step $j$ rather than the original BP gain. In $\pi_j$, the submodular part is down weighted relative to the supermodular part. Intuitively, this is helpful because the supermodular part is initially much smaller than the submodular part, but ultimately dominates the sum. Thus, it is in the optimizer's interest to focus on the supermodular part early, rather than waiting until it becomes large.

We define the **approximate distorted greedy** selection rule as follows. Given scalars $\{r_j\}_{j=1}^k$, in each step $j = \{1, \ldots, k\}$, the optimizer chooses an item $v_j$ that satisfies

$$v_j \in \{v : \pi_j(v|S_{j-1}) \geq \text{argmax}_{\tilde{v}} \, \pi_j(\tilde{v}|S_{j-1}) - r_j\}. \quad (4)$$

We present a robust version of Liu et al. [2022]:

**Lemma 3.** *Any output $S$ of the approximate distorted greedy selection rule in Equation* (4) *admits the following guarantee for Problem* (1) *with a BP objective (Def. 2):*

$$h(S) \geq \min\left\{1 - \frac{\kappa_f}{e}, \, 1 - \kappa^g\right\} h(S^*) - \sum_{j=1}^k r_j,$$

*where $\kappa_f, \kappa^g$ are as defined in Def. 3 and 4.*

This lemma is the key to the analysis of Algorithm 4 in Section 5. We remark that the approximation ratio above is different from Liu et al. [2022]. This is due to us fixing an error that we noticed in their analysis, which caused the approximation ratio to change from their $\alpha = \min\left\{1 - \frac{\kappa_{f,q}}{e}, \, 1 - \kappa_q^g e^{(1-\kappa_q^g)}\right\}$ to our above. Details are in Appendix H.1. Additionally, note that Sviridenko et al. [2017] provided a $1 - \frac{\kappa_f}{e}$ lower bound for monotone submodular maximization and later on, Bai and Bilmes [2018] obtained a $1 - \kappa^g$ lower bound for monotone supermodular maximization. Our approximation ratio in Equation (12) is simply the minimum of these two quantities. In Appendix H, we provide a heat map that compares this approximation ratio to that of Bai and Bilmes [2018], showing that it is strictly greater for all $\kappa_f, \kappa^g$. Once their analysis is fixed, we adapt their argument to the more general case that allows for errors $r_j$ at each stage to complete the robust online proofs.

## 4 NO-REGRET SINGLE FEEDBACK

In the previous section, we considered the robustness of the greedy algorithm in the offline setting. We now return to our interactive problem from Section 2 and will show how it reduces to the offline problem.

First, we fully define the notion of scaled regret mentioned in Section 1. The scaling is chosen to compare with the appropriate offline algorithm for the relevant function class; it is standard to consider scaled regret for NP-hard problems

(e.g., [Chen et al., 2017]). Recall our interactive setup from Section 2. Let $T_q$ represent the number of items selected for function $h_q$ by the final round, $T$, so that $\sum_{q=1}^m T_q = T$. The set $S_{T_q,q}$ is the final selection for $h_q$ and we set $S_q = S_{T_q,q}$ for notational simplicity. Let $S_q^* \in \text{argmax}_{|S| \leq T_q} h_q(S)$ be a maximizing payoff set for $h_q$ with at most $T_q$ elements. Inspired by Bai and Bilmes [2018] and with respect to the approximation ratio obtained for the greedy baseline for BP functions, we define the regret metric $\mathscr{R}_{\text{BP}}(T)$ as follows:

$$\mathscr{R}_{\text{BP}}(T) := \sum_{q=1}^m \frac{1}{\kappa_{q,f}} \left[1 - e^{-(1-\kappa_q^g)\kappa_{q,f}}\right] h_q(S_q^*) - h_q(S_q). \quad (5)$$

From Lemma 1, we recognize that if our online algorithm is approximately greedy as in Equation (2), then our regret will be bounded by the accumulation of the approximation errors $r_j$. This observation bridges the gap between the online and offline settings. Hence, our goal is to design an algorithm that satisfies this property. Analogously, for functions with bounded submodularity ratio $\gamma_q$ (Definition 5) and generalized curvature $\zeta_q$ (Definition 6), we define:

$$\mathscr{R}_{\text{WS}}(T) := \sum_{q=1}^m \frac{1}{\zeta_q} \left[1 - e^{-\zeta_q \gamma_q}\right] h_q(S_q^*) - h_q(S_q). \quad (6)$$

If we knew all the functions $\{h_1 \ldots h_m\}$, we could select the greedy item at each stage and achieve zero regret. Define $\Delta(\phi, S, v) = h_\phi(v|S)$ to encapsulate all our $m$ latent objectives succinctly; we also further below use the notational shortcuts $x_t = (\phi_{u_t}, S_{u_t}, v_t)$ and $\Delta(x_t)$. If we knew this function, we would know $\{h_1 \ldots h_m\}$ as well. Thus, our task is to design a procedure to estimate $\Delta(\phi, S, v)$ from data such that the approximation errors $r_j$ reduce over time.

To make this possible, we must make additional assumptions on $\Delta(\cdot)$. To see why, consider what we can infer from an observation without any additional assumptions. In the BP case, for instance, the $q$-th BP gain function is uniquely defined by $2^{|V|}$ function evaluations $h_q(v|S)$ for each possible $(v, S)$. If we observe $h_q(v|S)$ for some $(v, S)$, then we can only make inferences about $f_q(v|A)$ and $g_q(v|A)$ for all $A \subseteq S$ or $A \supseteq S$; since we can only choose item $v$ once during the optimization for user $q$, this information is not useful practically. This motivates the following assumption.[1]

**Assumption 1.** *The $\Delta(\cdot)$ function lives in a Reproducing Kernel Hilbert Space (RKHS) associated with some kernel $\hbar$ and has bounded norm i.e $\|\Delta\|_\hbar \leq B$.*

The assumption ensures the outputs of the $\Delta(\cdot)$ function vary smoothly with respect to the inputs and is standard with GPCBs [Seeger et al., 2008, Srinivas et al., 2010, Krause and Ong, 2011, Chen et al., 2017]. E.g., if two related movies are watched by two similar users, they should provide similar ratings. Thus, each query provides information about all

---

[1]See [Berlinet and Thomas-Agnan, 2011] for a comprehensive treatment of RKHS and kernels.

**Algorithm 1** MNN-UCB

**Input** set $V$, $\hbar$ kernel function
1: Init $S_q \leftarrow \varnothing, V_q \leftarrow V, \forall q \in [m]$
2: Init $X_0 \leftarrow \varnothing, G_1 \leftarrow \varnothing$
3: **for** $t \in 1, 2, 3 \ldots T$ **do**
4:     Observe $u_t$ from environment.
5:     **if** $t = 1$ **then**
6:         Choose $v_1 \in V_{u_t}$ uniformly at random
7:     **else**
8:         Update $\mathbf{y}_t \leftarrow [y_1, y_2, \ldots, y_{t-1}]^\top$
9:         $\mu_t, \sigma_t = \textsc{MeanVarCalc}$
10:           $(V_{u_t}, \phi_{u_t}, S_{u_t}, G_t, G_{t-1}, x_{t-1}, X_{t-1}, \mathbf{y}_t)$
11:         Select $v_t \leftarrow \arg\max_{v \in V_{u_t}} \mu_t(v) + \beta_t \sigma_t(v)$
12:     **end if**   // $\beta_t$ is defined in Eq. (27).
13:     Update $S_{u_t} \leftarrow S_{u_t} \cup \{v_t\}$,   $x_t \leftarrow (\phi_{u_t}, S_{u_t}, v_t)$, $X_t \leftarrow X_{t-1} \cup \{x_t\}, V_{u_t} \leftarrow V_{u_t} \setminus v_t$
14:     Obtain feedback $y_t = \Delta(x_t) + \epsilon_t$,
15:     // Decide whether to store new point.
16:     $G_{t+1} = \textsc{NyströmSelect}(\hbar, G_t, x_t)$
17: **end for**

---

**Algorithm 2** NyströmSelect Zenati et al. [2022].

**Input:** $\hbar, G_t, x_t$; **Locally stored variables:** List $L$
**Hyperparams:** Regularization $\lambda$, Accuracy $\eta$, Budget $b$
1: If first call, init $L$ to empty list.
2: Compute leverage score $\hat{\tau}_t(\lambda, \eta)$ from Eq. (7)
3: With probability $\min(b\hat{\tau}_t(\lambda, \eta), 1)$ include $x_t$ in $G_{t+1}$.
4: Append $\hat{\tau}_t(\lambda, \eta)$ to L
**Result:** $G_{t+1}$

---

$m \cdot 2^{|V|}$ other possible queries to all functions, making estimation feasible since the kernel $\hbar((\phi_q, S, v), (\phi_{q'}, S', v'))$ measures similarity between two inputs.

### 4.1 MNN-UCB ALGORITHM

Algorithm 1 is inspired by [Chen et al., 2017, Zenati et al., 2022] based on Upper Confidence Bound (UCB) algorithms for kernel bandits. At time $t \in [T]$, the optimizer has available the noisy evaluations of the unknown $\Delta(\cdot)$ function in vector $\mathbf{y}_t = (y_j)_{j=1}^{t-1}$ for corresponding inputs held in vector $X_t = (x_j)_{j=1}^{t-1}$ — these are updated at the end of each iteration. These are used by the subroutine MVCALC that, using GP kernel techniques [Valko et al., 2013, Srinivas et al., 2010] and the Nyström set of samples $G_t \subset X_t$, efficiently compute estimates of the GP posterior distribution's conditional mean and variance used for the UCB marginal gains in the maximization (line 10 of Algorithm 1).

That is, the algorithm chooses the item $v_t$ that has the highest UCB in line 11 where the parameter $\beta_t$ controls the algorithm's propensity towards either exploration or exploitation (see Appendix G.2). We use the notation $\hbar_A(x) = [\hbar(x_1, x) \ldots \hbar(x_{|A|}, x)]$ to measure the similar-

**Algorithm 3** MeanVarCalc.

**Input:** $V_{u_t}, \phi_{u_t}, S_{u_t}, G_t, G_{t-1}, x_{t-1}, S, \mathbf{y}_t$
**Locally stored variables:** $L_1, L_2, L_3$
**Hyperparams:** Regularization $\lambda$
1: // Update $K_{GG}^{-1}, \Lambda_t, \tilde{\mathbf{y}}_t$.
2: **if** $|G_t| = 1$ **then**
3:     Init $L_1, L_2, L_3$ each to empty lists
4:     Init $\tilde{\mathbf{y}}_t = y_t \hbar(x_{t-1}, x_{t-1})$
5:     Init $K_{G_t G_t}^{-1} = 1/\hbar(x_{t-1}, x_{t-1})$
6:     Init $\Lambda_t = 1/[\hbar(x_{t-1}, x_{t-1})^2 + \lambda \hbar(x_{t-1}, x_{t-1})]$.
7: **else if** $G_t = G_{t-1}$ **then**
8:     Update $\Lambda_t$ using Eq (8)
9:     $\tilde{\mathbf{y}}_t = \tilde{\mathbf{y}}_{t-1} + y_t \hbar_{G_t}(x_{t-1})$
10: **else**
11:     Update $\Lambda_t$ using Eq. (8) with Schur complements.
12:     Update $K_{G_t G_t}^{-1}$ using Eq (10) with Schur complements // here and line 11 use lists $L_1, L_2, L_3$
13:     $\tilde{\mathbf{y}}_t = [\tilde{\mathbf{y}}_{t-1} + y_t \hbar_{G_t}(x_{t-1}), K_S(x_{t-1})^\top \mathbf{y}_t]^\top$
14: **end if**
15: $z_t \leftarrow (\phi_{u_t}, S_{u_t})$
16: Append $(K_{GG}^{-1}, \Lambda_t, \tilde{\mathbf{y}}_t)$ to $L_1, L_2, L_3$ lists resp
17: // Calculate mean and variance vectors.
18: **for** $v \in V_{u_t}$ **do**
19:     $\tilde{\mu}_t(v) \leftarrow \hbar_{G_t}((z_t, v))^T \Lambda_t \tilde{\mathbf{y}}_t$
20:     $\delta_t(v) \leftarrow \hbar_{G_t}((z_t, v))^T (\Lambda_t - \lambda^{-1} K_{GG}^{-1}) \hbar_{G_t}((z_t, v))$
21:     $\tilde{\sigma}_t(v)^2 \leftarrow \lambda^{-1} \hbar((z_t, v), (z_t, v)) + \delta_t(v)$
22: **end for**
**Result:** $\{\tilde{\mu}_t(v)\}_{v \in V_{u_t}}, \{\tilde{\sigma}_t(v)\}_{v \in V_{u_t}}$

---

ity between $x$ and every element in $A = \{x_j\}_{j \in \{1, \ldots, |A|\}}$. Hence, $\hbar_{G_t}((\phi_{u_t}, S_{u_t}, v))$ measures the similarity of the input $(\phi_{u_t}, S_{u_t}, v)$ to the historical data in Nyström set $G_t$. Notation $K_{AB}(v) = [\hbar(x, x')]_{x \in A, x' \in B}$ contains the matrix of pairwise kernel-similarities for elements in $A$, $B$ and $K_{G_t G_t}$ is the covariance matrix of the historical data $G_t$. Below, we describe the details for the two subroutines used in Algorithm 1, for the readers who are interested in calculations for kernel updates, and selection of informative points to improve computation via Nyström sampling.

**Efficiency and NyströmSelect** In prior submodular bandits work [Chen et al., 2017], each iteration $t \in [T]$ needs to invert a $t \times t$ matrix since *all* historical data $X_t$ is used when calculating the conditional means and variances. Even if online matrix-inverse techniques are used, the runtime becomes $O(T^3)$, which is impractical. We use Nyström sampling to mitigate this and only use a selected subset $G_t \subset X_t$ of historical data to compute $\mu_t(v), \sigma_t(v)$ for all $v \in V_{u_t}$ [Zenati et al., 2022]. Nyström sampling chooses the points that are most useful for prediction. To define this precisely, we introduce a bit of notation. Define $G' = G_t \cup x_t$. Define the estimated leverage score $\hat{\tau}_t(\lambda, \eta, x)$ as:

$$\frac{1 + \eta}{\lambda} \left[ \hbar(x, x) - \tilde{\hbar}_{G'}(x)(\tilde{K}_{G'G'} + \lambda I)^{-1} \tilde{\hbar}_{G'}(x) \right] \quad (7)$$

Define:
$$M_t = \begin{bmatrix} \text{diag}\left([\min(\hat{\tau}_j(\lambda, \eta, x_j), 1)]_{x_j \in G_t}\right) & \mathbf{0} \\ \mathbf{0} & 1 \end{bmatrix}$$

It is the diagonal scaling matrix of (clipped) leverage scores of past selected points with an extra entry with value 1 for the hypothetical new point $x_t$. In Equation (7) above, we define $\tilde{k}_{G'}(x_t) = M_t^\top k_{G'}(x_t)$ and $\tilde{K}_{G'G'} = M_t^\top K_{G'G'} M_t$. The point $x_t$ is included into the Nyström set $G_{t+1}$ in with probability proportional to $\hat{\tau}_t(\lambda, \eta, x_t)$ (line 3 of Algorithm 2). To understand why this is reasonable, note that $\hat{\tau}_t(\lambda, \eta, x_t)$ is shown to estimate the ridge leverage score (RLS) of a point well [Calandriello et al., 2017, 2019]. The RLS measures intuitively how correlated the new point is to previous points; if it is highly correlated, it will be sampled with low probability, but if it is orthogonal, it will be sampled with high probability. This procedure improves the runtime of Algorithm 1 to $O(T|G_T|^2)$, where $|G_T|$ is the number of selected Nyström points until the final timestep. A discussion on setting the hyperparameters $\eta$ and $b$, controlling the tradeoff between regret and computation, is given in Appendix G.1.

**MVCALC**  This subroutine calculates the posterior mean and variance for $\Delta(\cdot)$ using Gaussian process posterior calculations after projecting on the Nyström points $G_t$. We define the intermediate quantity
$$\Lambda_t = (K_{G_t S_t} K_{S_t G_t} + \lambda K_{G_t G_t})^{-1},$$
which is useful in these updates. Note that the algorithms store and track the local variables $K_{GG}^{-1}, \Lambda_t, \tilde{\mathbf{y}}_t$ across time steps. It needs to incrementally invert $K_{GG}^{-1}$ and $\Lambda_t$ as the time steps continue. For $\Lambda_t$, if $G_t$ does not change, it does this using the Sherman-Morrison formula:
$$\Lambda_t = \Lambda_{t-1} - \frac{\Lambda_{t-1} k_{G_t}(x_t) k_{G_t}(x_t)^\top \Lambda_{t-1}}{1 + k_{G_t}(x_t)^\top \Lambda_{t-1} k_{G_t}(x_t)}. \quad (8)$$
This update takes $|G_t|^2$ time. In the case that $G_t$ changes, we can write the expression for $\Lambda_{t+1}$ as follows. In the below, let $a = K_{G_t}(x_t)$ and $c = k(x_t, x_t)$.
$$\Lambda_{t+1} = \begin{bmatrix} K_{G_t S} K_{S G_t} + aa^\top & K_{G_t S}^\top a + ca \\ a^\top K_{S G_t} + ca & a^\top a + c^2 \end{bmatrix}^{-1}. \quad (9)$$
We can use the Schur complement block-matrix inverse identity to evaluate the above which takes $O(t|G_t|)$ time. Similarly, for $K_{G_t G_t}^{-1}$, we write it in block-matrix form as:
$$K_{G_t G_t}^{-1} = \begin{bmatrix} K_{G_{t-1} G_{t-1}} & K_{G_{t-1}}(x_t) \\ K_{G_{t-1}}(x_t)^\top & k(x_t, x_t) \end{bmatrix}^{-1}, \quad (10)$$
computable using Schur complements in $|G_t|^2$ time.

## 4.2  THEORETICAL GUARANTEE

For a given set $X_T = \{x_1, \dots, x_T\}$, all our bounds are stated in terms of the effective dimension of the matrix $K_T = K_{X_T X_T}$, as described in Hastie et al. [2009]:

**Definition 1.** $d_{eff}(\lambda, T) = Tr(K_T(K_T + \lambda I)^{-1})$

Intuitively, the effective dimension is a measure of the number of dimensions in the feature space that are needed to capture data variations. Having a smaller effective dimension enables learning the unknown $h_q$ with fewer samples. If the empirical kernel matrix $K_T$ has eigenvalues $(\lambda_1 \dots \lambda_T)$, the effective dimension can equivalently be written as $d_{eff}(\lambda, T) = \sum_{t=1}^T \frac{\lambda_t}{\lambda_t + \lambda}$. Thus, if the eigenvalues decay quickly, the denominator will dominate for most summands and $d_{eff}$ will be small. If the kernel is finite dimensional, with dimension $s$, then only the first $s$ terms in the summation will be nonzero and $d_{eff} \leq s$. Having a small effective dimension makes the problem of learning the unknown objective function easier and hence improves our regret guarantee. This quantity is inspired by classical work in statistics [Hastie et al., 2009, Zhang, 2002].

It is instructive to bound $d_{eff}$ for different kernels. We relate $d_{eff}$ to the information gain $\tilde{\gamma}(\lambda, T)$ [Srinivas et al., 2010].
$$d_{eff}(\lambda, T) = \sum_{t=1}^T \frac{\frac{\lambda_t}{\lambda}}{\frac{\lambda_t}{\lambda} + 1} \leq \sum_{t=1}^T \log\left(1 + \frac{\lambda_t}{\lambda}\right) = \tilde{\gamma}(\lambda, T).$$
Here, we used the inequality $\frac{x}{x+1} \leq \log(1 + x)$ for $x \geq -1$. Srinivas et al. [2010] provides bounds on $\tilde{\gamma}(\lambda, T)$ for various kernels. For the (most popular) Gaussian kernel with dimension $d$, they show that $\tilde{\gamma}(\lambda, T) \leq \log(T)^{d+1}$ which holds under the assumption that the eigenvalues $\lambda_t$ are square summable. We plot the exact $d_{eff}(\lambda, T)$ for the Gaussian kernel in Figure 3 thereby verifying this bound empirically. For the linear kernel with dimension $d$, we have $\tilde{\gamma}(\lambda, T) \leq d \log(T)$. In our experimental results (see Section I), we use a composite over three constituent kernels and Theorems 2 and 3 of Krause and Ong [2011] bound $d_{eff}$ for the product or sums of kernels, when $\tilde{\gamma}(\lambda, T)$ is bounded for each constituent. This all ensures our regret bounds are sublinear in practice.

Now, we are ready to state our main result.

**Theorem 1.** *Let Assumption 1 hold and assume that $\epsilon_t$ are i.i.d centered sub-Gaussian (i.e., light tailed) noise. Then MNN-UCB (Algorithm 1) obtains the following regret:*
*(a) When all $h_q$ are BP functions, we have that*
$$\mathbb{E}[\mathcal{R}_{BP}(T)] \leq O\left(\sqrt{T}\left(B\sqrt{\lambda d_{eff}} + d_{eff}\right)\right)$$
*(b) When all $h_q$ are WS functions, we have that*
$$\mathbb{E}[\mathcal{R}_{WS}(T)] \leq O\left(\sqrt{T}\left(B\sqrt{\lambda d_{eff}} + d_{eff}\right)\right)$$

Below, we prove case (a); for the proof of case (b), refer to Appendix G. We see that Lemma 1 and our algorithm design enables us to relate our notion of regret with the pointwise notion of regret from Zenati et al. [2022].

*Proof.* We define $R_t = \sum_{j=1}^t r_j$ where $r_t = \sup_{v \in V} h_{u_t}(v|S_{t_{u_t}}, u_t) - h_{u_t}(v_t|S_{t_{u_t}}, u_t)$. Notice $R_t$ is different from $\mathcal{R}_{BP}(t)$. From Lemma 1 for all $q$, and with $\mathcal{R}_{BP}(T)$ defined in Equation (5), we have $\mathcal{R}_{BP}(T) \leq \sum_{t=1}^T r_t = R_T$. We model the problem of the present work

as a contextual bandit problem in the vein of Zenati et al. [2022]. Here, the context in round $t$ is $z_t = (\phi_{u_t}, S_{t_{u_t}, u_t})$. We next invoke Theorem 4.1 in Zenati et al. [2022] to complete our result. Further details are in Appendix G. □

# 5 NO-REGRET SEPARATE FEEDBACK

In the previous section, for BP functions, we obtained sublinear $\alpha$-regret with respect to the offline greedy baseline under the *monolithic feedback* setting from Section 2. In this section, we study whether obtaining *seperate feedback* for the submodular and supermodular parts of the BP function can enable us to provide a stronger guarantee. Towards, this end, we make the following assumption:

**Assumption 2.** *For any $(v, A)$, the optimizer has access to two oracles that provide it with separate feedback for the submodular $f_q(v|A)$ and supermodular $g_q(v|A)$ parts.*

This would be satisfied for applications where we have more fine-grained feedback than just a single reward. For movie recommendations, we could ask users questions like "why did you like this movie?" or "what would your rating for this movie have likely looked like if you had not watched X?". Alternatively, we could estimate this from other similar users who had not watched "X" – this is likely to be what would be done in practice when a large user base is available. For training subset selection, for any chosen point, we could look at the density of nearby points from the opposite class to deduce whether the benefit is coming from complementary selection of points - see Figure 2 for an illustration.

In addition to the above, we need another technical assumption. In order to state this, we first introduce some notation. As in Section 3, we define for each function $q$ the modular lower bound $l_{q,1}(A)$ as the totally normalized submodular function $f_{q,1}$ and also $\pi_{j,\phi_q}(v|A)$ as:

$$\left(1 - \frac{1}{T_q}\right)^{T_q - j - 1} f_{q,1}(v|A) + g_q(v|A) + l_{q,1}(v). \quad (11)$$

We use $\pi_{j,q} = \pi_{j,\phi_q}$ interchangeably for readability. Now, we additionally assume the following for each $q \in [m]$.

**Assumption 3.** *(a) The modular lower bound $l_{q,1}(\cdot)$ is known by the optimizer. (b) The number of items for each user $T_q$ is known by the optimizer.*

For Assumption (3a), note the modular lower bound is the summation over items for the minimum possible submodular gain of selecting that item. For the case of movie recommendation, there is likely to be some marginal enjoyment from watching a movie, even if all other movies in the set have already been watched; this would of course depend on the overall size of the ground set. We could imagine that domain knowledge could indicate what these "least gain"

quantities would look like - if we are unsure, we can always choose a conservative estimate and the bound would degrade smoothly. Note also that the modular lower bound is defined by $|V|$ function evaluations, whereas the submodular function is defined by $2^{|V|}$. Hence, the submodular $f_q$ is still mostly unknown as knowing $l_{q,1}$ is a much weaker assumption than the offline setting. For Assumption (3b), if $T_q$ is not known beforehand, "guess and double" techniques (Appendix H.4) can be used, the effects of which result in a bounded additive term.

Given the richer feedback model, we provide sublinear regret guarantees for the stronger notion of regret for BP functions with respect to the **distorted greedy** baseline (Section 3.2). This is defined as:

$$\mathcal{R}_{\text{BP},2}(T) :=$$
$$\sum_{q=1}^{m} \min\left\{1 - \frac{\kappa_{f,q}}{e}, \ 1 - \kappa_q^g\right\} h_q(S_q^*) - h_q(S_q). \quad (12)$$

For applications where Assumption (3a) is not satisfied, we additionally provide an alternate result in Appendix H.3 without this assumption. Here, the $\alpha$ is slightly reduced to $\min\left\{1 - \frac{1}{e}, \ 1 - \kappa_q^g\right\}$. However, the heat map in Figure 4 illustrates the bounds are still better than the vanilla greedy $\alpha$ from Bai and Bilmes [2018] for most choices of $\kappa_{f,q}, \kappa_q^g$. And in cases where Assumption 2 does not hold, we can default to the results from Section 4.2.

## 5.1 ALGORITHM AND THEORETICAL GUARANTEE

We present below our modified algorithm for the case of richer feedback.

---
**Algorithm 4** MNN-UCB-Separate (modified Algorithm 1)
Line 14 of Algorithm 1 replaced with the following:
1: Calculate distortion $D_t \leftarrow (1 - \frac{1}{T_{u_t}})^{T_{u_t} - |S_{u_t}| - 1}$.
2: Obtain submodular feedback $y_{f,t} = f_{u_t}(v_t|S_{u_t}) + \epsilon_{f,t}/2$ and $y_{g,t} = g_{u_t}(v_t|S_{u_t}) + \epsilon_{g,t}/2$.
3: Apply distortion to obtain overall feedback $y_t = D_t y_{f,t} + y_{g,t} + (1 - D_t)l_{u_t,1}(v_t)$.

---

Algorithm 4 is quite similar to Algorithm 1 — line 14 of Algorithm 1 is modified to the three steps of Algorithm 4. That is, the feedback, now obtained separately as $y_{f,t}$ for the submodular and as $y_{g,t}$ for the supermodular part, is aggregated in line 3 of Algorithm 4 as per $\pi_{j,\phi_q}$. Our guarantee for Algorithm 4 is the following:

**Theorem 2.** *Let Assumptions 1, 2, 3 hold and assume that $\epsilon_t$ are i.i.d centered sub-Gaussian noise. Then, when all $h_q$ are BP functions, Algorithm 4 yields*

$$\mathbb{E}[\mathcal{R}_{BP,2}(T)] \leq O\left(\sqrt{T}\left(B\sqrt{\lambda d_{\text{eff}}} + d_{\text{eff}}\right)\right)$$

The proof follows along similar lines as Theorem 1 and is included in Appendix H.2.

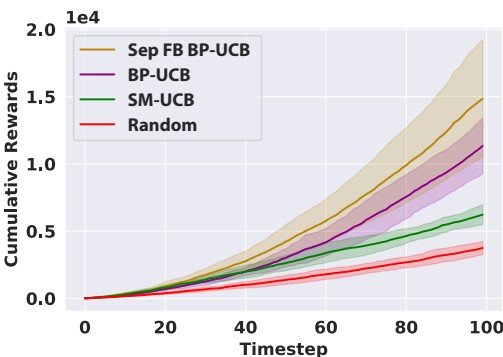

Figure 1: Algorithm 1 (magenta, green) and Algorithm 4 (gold) applied to the MovieLens dataset. The highlighted region shows the standard deviation over 10 random trials.

# 6 NUMERICAL EXPERIMENTS

From MovieLens, we obtain a ratings matrix $M \in \mathbb{R}^{900 \times 1600}$, where $M_{i,j}$ is the rating of the $i^{\text{th}}$ user for the $j^{\text{th}}$ movie. Using this dataset, we instantiate an interactive BP maximization problem, as formulated in Vignette 1. We cluster the users into $m = 10$ groups using the $k$-means algorithm and design a BP objective for each user-group. The objective for the $q_{th}$ group is decomposed as $h_q(A) = \sum_{v \in A} m_q(v) + \lambda_1 f_q(A) + \lambda_2 g_q(A)$, where the modular part $m_q(v)$ is the average rating for movie $v$ amongst all users in group $k$. The concave-over-modular submodular part encourages the recommender to maintain a balance across genres in chosen suggestions. By contrast, the supermodular function is designed to encourage the optimizer to exploit complementarities within genres. The constants $\lambda_1, \lambda_2$ are chosen so that the supermodular part slightly dominates the submodular part, since previous work already studies primarily submodular functions. Further details are provided in Section I. Note that these functions are hand-crafted for illustration purposes and are unknown to the optimizer in all cases.

In Figure 1, the red curve corresponds to a naive baseline where movies are recommended at random. The green curve corresponds to the the algorithm from Chen et al. [2017], where the supermodular part is ignored. The magenta curve corresponds to Algorithm 1 from Section 4 in the case where monolithic feedback is provided to the optimizer. The gold curve corresponds to Algorithm 4 when separate feedback is provided to the optimizer.

Since both the gold and purple curves are significantly better than the green, we see the pitfall of modeling a BP problem as purely submodular. The improved performance of the gold curve over the purple shows the impact of the stronger feedback in Assumptions 2, 3 on performance. This corroborates that the regret guarantee in Theorem 2 is stronger than that from Theorem 1. Active learning experiments are given in Appendix D.1.

# 7 DISCUSSION

In this paper, we presented algorithms for efficient online optimization of certain non-submodular functions, which enables better modeling in some applications. We considered two different feedback models and provided variants of optimistic kernel-bandit algorithms that achieve sublinear regret. Along the way, we studied the robustness of the greedy algorithm for these function classes in Section 3, which is of independent interest.

**Limitations and Future Work.** A limitation of Gaussian Process based methods is that the model updates at each stage are computationally expensive, making them difficult to use in practice. While we made headway towards addressing this problem, our time complexity of $O(T|G_T|^2)$ may still be prohibitive for some applications of optimizing submodular or beyond submodular functions. In these cases, other prediction and uncertainty quantification techniques may be used - such as the bootstrap. Even simpler heuristic approaches may be employed that balance between areas of the input space that the learned model thinks are promising, and those that are underexplored. Our theory suggests that such methods are likely to work well.

In general, a decomposition of the utility function $h$ could take other forms, such as a quotient of two submodular functions, or as a product of a submodular and supermodular function. The additive BP form is emphasized due to its analytical tractability, its ability to understand bounds, and its greater expressivity, but the future holds no limit regarding possible decomposable $h$s that are viable in the separate feedback setting that we introduced. A theoretical artifact of the curvature-based bound for non-modular functions is that they are independent of $f_q$ and $g_q$'s relative magnitude which are important for experiments. It would be useful to develop guarantees that incorporate this.

**Applications and Social Impact.** In Appendix D, we compare our proposed approach with others from the literature for the motivating applications of recommendation systems and active learning. We include some illustrative examples, which show the utility of modeling these problems as BP instead of purely submodular.

For recommendation systems, we conjecture that explicitly modeling the diminishing/increasing returns of utilities could help alleviate the primary problem of state-of-the-art systems today - that they are more likely to show addictive and harmful content in order to keep users glued onto the service. For active learning, we conjecture that considering utilities beyond submodular is likely to provide us with more flexibility in expressing and balancing our multiple goals when choosing a subset of datapoints for training.

**Acknowledgements**  AN would like to thank Mitas Ray for valuable early conversations that led to the inception of this project. In addition, we would like to thank the anonymous reviewers for their helpful feedback. While working on this project, AN was supported by the Amazon Hub Fellowship at the University of Washington. This work was also supported in part by NSF TRIPODS II-DMS 20231660, NSF CCF 2212261, NSF CCF 2007036, NSF AF 2312775, NSF IIS-2106937, NSF IIS-2148367.

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

# A    TABLE OF NOTATION

| Notation | Description |
|---|---|
| $V$ | Ground set of items |
| $m$ | Number of set functions |
| $h_q$ | $q$-th set function, $q \in [m]$ |
| $u_t$ | Index of arrived function at time $t$ |
| $\phi_{u_t}$ | Context vector for function $h_{u_t}$ at time $t$ |
| $v_t$ | Item selected at time $t$ |
| $S_{k,q}$ | Items selected for function $h_q$ up to time $k$ |
| $y_t$ | Noisy marginal gain feedback at time $t$ |
| $y_{f,t}, y_{g,t}$ | Separate submodular and supermodular feedback at time $t$ |
| $S_q^*$ | Optimal set for function $h_q$ |
| $T_q$ | Number of items selected for function $h_q$ by time $T$ |
| $\kappa_f, \kappa^g$ | Submodular and supermodular curvatures |
| $\gamma, \zeta$ | Submodularity ratio and generalized curvature |
| $\mathscr{R}_{\mathrm{BP}}(T), \mathscr{R}_{\mathrm{WS}}(T)$ | Regret for BP and WS functions |
| $\mathscr{R}_{\mathrm{BP},2}(T)$ | Regret for BP functions with separate feedback |
| $\Delta(\phi, S, v)$ | Marginal gain of adding $v$ to $S$ for context $\phi$ |
| $\mathscr{K}$ | Reproducing kernel Hilbert space (RKHS) |
| $B$ | Bound on RKHS norm of $\Delta$ |
| $G_t$ | Nyström set at time $t$ |
| $\beta_t$ | Exploration-exploitation tradeoff parameter |
| $d_{\mathrm{eff}}(\lambda, T)$ | Effective dimension |
| $l_1, l_2$ | Modular lower bounds for $f$ and $g$ |
| $f_1, g_1$ | Totally normalized $f$ and $g$ |
| $\pi_j(v|A)$ | Distorted marginal gain for selecting $v$ given $A$ at step $j$ |

Table 2: Table of key notation used in the paper.

# B    BACKGROUND ON SUBMODULARITY, SUPERMODULARITY AND CURVATURES

A set function $h : 2^V \to \mathbb{R}$ is a function that maps any subset of a finite ground set $V$ of size $|V| = n$ to the reals. There are many possible set functions, and arbitrary set functions are impossible to optimize with any quality assurance guarantee without an exponential cost. As an example, consider a function $h$ such that $h(A) = a > b$ for some set $A \subseteq V$ and $a, b \in \mathbb{R}$ and $h(B) = b > 0$ for all $B \neq A$. Then any algorithm that does not search over all $2^{|V|}$ subsets can miss set $A$ and the approximation ratio $a/b$ can be unboundedly large. We are therefore interested in set functions that have useful and widely applicable structural properties such as the class of submodular and supermodular functions.

A set function $f : 2^V \to \mathbb{R}$ is said to be monotone non-decreasing if $f(A \cup \{v\}) \geq f(A)$ for all $A \subseteq V, v \in V$. It is normalized if $f(\emptyset) = 0$. For convenience, we refer to the collection of **Monotone Non-decreasing Normalized** set functions as **MNN** functions. We use the gain notation $f(v|S) = f(S \cup \{v\}) - f(S)$ to denote the **marginal gain** of adding element $v$ to the set $S$.

A set function $f$ defined over the ground set $V$ is called **submodular** if for all $A \subseteq B \subseteq V$ and any element $v \notin B$ we have $f(A \cup \{v\}) - f(A) \geq f(B \cup \{v\}) - f(B)$. A function $g : 2^V \to R$ is said to be **supermodular** if $-g$ is submodular — $g$ has the property of *increasing returns* where the presence of an item can only enhance the utility of selecting another item. The class of functions defined below is the primary focus of our paper.

**Definition 2** (BP Function)**.** *A utility function $h$ is said to be BP if it admits the decomposition $h = f + g$, where $f$ is submodular, $g$ is supermodular, and both functions are also MNN.*

Next, we introduce the notion of curvature for submodular and supermodular functions. This will enable us to understand the assumptions required to obtain approximation bounds for offline BP functions, as stated in Bai and Bilmes [2018].

**Definition 3** (Submodular curvature). *Denote the curvature for submodular $f$ as $\kappa_f = 1 - \min_{v \in V} \frac{f(v|V \setminus \{v\})}{f(v)}$.*

**Definition 4** (Supermodular curvature). *Denote the curvature for supermodular $g$ as: $\kappa^g = 1 - \min_{v \in V} \frac{g(v)}{g(v|V \setminus \{v\})}$.*

These quantities are contained in $[0, 1]$ and measure how far the functions are from being modular: if a curvature is zero, the function is modular. Important for practical applications, given the function, these can be calculated in linear time in $|V|$. Bai and Bilmes [2018] analyzed the greedy algorithm for the cardinality-constrained BP maximization problem and provided a $\frac{1}{\kappa_f} \left[ 1 - e^{-(1-\kappa^g)\kappa_f} \right]$ approximation ratio for this problem. They also showed that not all monotone non-decreasing set functions admit a BP decomposition. However, in cases where such a decomposition is available, one can easily compute the curvature of submodular and supermodular terms and compute the bound.

Since not all MNN functions are representable as BP functions, we also study arbitrary MNN functions in terms of how far they are from being submodular.

**Definition 5** (Submodularity ratio, [Bian et al., 2019, Das and Kempe, 2011, Calandriello et al., 2018]). *The submodularity ratio of a non-negative set function $h(\cdot)$ is the largest scalar $\gamma$ such that $\sum_{v \in S \setminus A} h(v|A) \geq \gamma h(S|A), \forall S, A \subseteq V$.*

The submodularity ratio measures to what extent $h(\cdot)$ has submodular properties. For a non-decreasing function $h(\cdot)$, it holds that $\gamma \in [0, 1]$ always, and $h(\cdot)$ is submodular if and only if $\gamma = 1$.

**Definition 6** (Generalized curvature, [Bian et al., 2019]). *The curvature of a non-negative function $h(\cdot)$ is the smallest scalar $\zeta$ such that $\forall S, A \subseteq V, v \in A \setminus S, h(v|A \setminus \{v\} \cup S)) \geq (1 - \zeta)h(v|A \setminus \{v\})$.*

Note that unlike the notions of submodular and supermodular curvature, the submodularity ratio and generalized curvature parameters are information theoretically hard to compute in general [Bai and Bilmes, 2018]. We refer to the MNN set functions with bounded submodularity ratio $\gamma$ and generalized curvature $\zeta$ as **weakly submodular** (WS). Bian et al. [2019] analyzed the greedy algorithm for maximizing such functions subject to a cardinality constraint and obtained a $\frac{1}{\zeta}\left(1 - e^{-\zeta\gamma}\right)$ approximation ratio for this problem.

## C  OTHER RELATED WORK

**Submodular maximization with bounded curvature.**  Nemhauser et al. [1978] studied the performance of the greedy algorithm for maximizing a monotone non-decreasing submodular set function subject to a cardinality constraint and provided a $1 - \frac{1}{e}$ approximation ratio for this problem. While Nemhauser and Wolsey [1978] showed that the $1 - \frac{1}{e}$ factor cannot be improved under polynomial number of function value queries, the performance of the greedy algorithm is usually closer to the optimum in practice. In order to theoretically quantify this phenomenon, Conforti and Cornu'ejols [1984] introduced the notion of *curvature* $\kappa \in [0, 1]$ for submodular functions—this is defined in Section B. The constant $\kappa$ measures how far the function is from being modular. The case $\kappa = 0$ corresponds to modular functions and larger $\kappa$ indicates that the function is more curved. Conforti and Cornu'ejols [1984] showed the greedy algorithm applied to monotone non-decreasing submodular maximization subject to a cardinality constraint has a $\frac{1}{\kappa}(1 - e^{-\kappa})$ approximation ratio. Therefore, for general submodular functions ($\kappa = 1$), the same $1 - \frac{1}{e}$ approximation ratio is obtained. However, if $\kappa < 1$, $\frac{1}{\kappa}(1 - e^{-\kappa}) > 1 - \frac{1}{e}$ holds and as $\kappa \to 0$, the approximation ratio tends to 1. More recently, Sviridenko et al. [2017] proposed two approximation algorithms for the more general problem of monotone non-decreasing submodular maximization subject to a matroid constraint and obtained a $1 - \frac{\kappa}{e}$ approximation ratio for these two algorithms. They also provided matching upper bounds for this problem showing that the $1 - \frac{\kappa}{e}$ approximation ratio is indeed optimal. Later on, the notion of curvature was extended to continuous submodular functions as well and similar bounds were derived for the maximization problem [Sadeghi and Fazel, 2021a, 2020, 2021b, Sessa et al., 2019].

**BP maximization**  Bai and Bilmes [2018] first introduced the problem of maximizing a BP function $h = f + g$ (Definition 2) subject to a cardinality constraint as well as the intersection-of-$p$-matroids constraint. They showed that this problem is NP-hard to approximate to any factor without further assumptions. However, if the supermodular function $g$ has a bounded curvature (i.e., $\kappa^g < 1$), it is possible to obtain approximation ratios for this problem. In particular, for the setting with a cardinality constraint, they analyzed the greedy algorithm along with a new algorithm (SemiGrad) and provided a $\frac{1}{\kappa_f}\left(1 - e^{-(1-\kappa^g)\kappa_f}\right)$ approximation ratio. Note that for general supermodular functions ($\kappa^g = 1$), the approximation ratio is 0 and as $\kappa^g \to 0$, the bound tends to that of Conforti and Cornu'ejols [1984] for monotone non-decreasing submodular maximization subject to a cardinality constraint. Bai and Bilmes [2018] also showed that not all monotone non-decreasing

set functions admit a BP decomposition. However, in cases where such a decomposition is available, one can compute the curvature of submodular and supermodular terms in linear time and compute the bound. More recently, Liu et al. [2022] proposed a distorted version of the greedy algorithm for this problem and provided an improved $\min\{1-\frac{\kappa_f}{e}, 1-\kappa^g e^{-(1-\kappa^g)}\}$ approximation ratio Liu et al. [2022].

**Submodularity ratio.** Das and Kempe [2011] introduced the notions of submodularity ratio $\gamma$ and generalized curvature $\zeta$ for general monotone non-decreasing set functions (defined in Section B) and showed that the greedy algorithm obtains the approximation ratio $\frac{1}{\zeta}(1-e^{-\zeta\gamma})$ under cardinality constraints [Bian et al., 2019, Das and Kempe, 2011]. Unlike the BP decomposition, the notions of submodularity ratio and generalized curvature can be defined for any monotone non-decreasing set function but is, in general, exponential cost to compute.

**Adaptive and Interactive Submodularity.** Chen et al. [2017] is the work most related to ours – they employ a similar UCB algorithm to optimize an unknown submodular function in an interactive setting. They define regret as the sub-optimality gap with respect to a full-knowledge greedy strategy at the final round. They define a different notion of pointwise regret as the difference between the algorithm's rewards and that of the greedy strategy *at that stage*, treating the past choices as fixed. By viewing the submodular problem as a special case of contextual bandits, they observe that this accumulation of pointwise regret is precisely bounded by Krause and Ong [2011]. Then, they modify the seminal proof in Nemhauser et al. [1978] to relate their target notion of regret with the pointwise regrets. Guillory and Bilmes [2010], Golovin and Krause [2011] also consider different adaptive or interactive submodular problems. They both assume more knowledge of the structure of the submodular objective than Chen et al. [2017].

**Kernel bandits.** Srinivas et al. [2010] consider the problem of optimizing an unknown function $f$ that is either sampled from a Gaussian process or has bounded RKHS norm. They develop an upper-confidence bound approach, called GP-UCB, that achieves sublinear regret with respect to the optimal, which depends linearly on an information-gain term $\gamma_T$. Krause and Ong [2011] extend the setting from Srinivas et al. [2010] to the contextual setting where the function $f_{z_t}$ being optimized now depends also on a context $z_t$ that varies with time. Valko et al. [2013] replace the $\gamma_T$ scaling with $\sqrt{\gamma_T}$. Camilleri et al. [2021] extend experimental design for linear bandits to the kernel bandit setting, and provide a new analysis, which also incorporates batches. Zenati et al. [2022] use Nyström points to speed up the algorithm, with the same asymptotic regret guarantee as GP-UCB, which inspires the algorithms developed in this paper.

**Combinatorial Bandits** In [Takemori et al., 2020, Papadigenopoulos and Caramanis, 2021, Kveton et al., 2014, Chen et al., 2018], the optimizer chooses a set at each time step and the submodularity is between these elements chosen in the single time step. In our work, the optimizer chooses a single item at each time and accumulates a set over time; the submodularity is between these elements chosen at different time steps. While the formulations are similar, the formulation in our paper would apply to different applications than this body of work.

**Comparison with [Liu et al., 2021, Chen et al., 2020]** These papers have titles similar to the present work, but actually apply to a very different setting. Consider Algorithm 1 from [Liu et al., 2021], which describes the setting they work within. This algorithm is reproduced below for ease of reference. We remark that the setting is better described as "streaming" rather

---

**Algorithm 5** [Liu et al., 2021]: Greedy for "online" (streaming) BP maximization

1: $S_0 \leftarrow \emptyset$
2: **for** each element $u_t$ revealed **do**
3:     **if** $t < k$ **then**
4:         $S_t \leftarrow S_{t-1} + u_t$
5:     **else**
6:         Let $u_j'$ be the element of $S_{t-1}$ maximizing $h(S_{t-1} + u_t - u_j')$
7:         **if** $h(S_{t-1} + u_t - u_j') - h(S_{t-1}) > \frac{c(h(S_{t-1}))}{k(1-\epsilon)}$ **then**
8:             $S_t \leftarrow S_{t-1} + u_t - u_j'$
9:         **else**
10:             $S_t \leftarrow S_{t-1}$
11:         **end if**
12:     **end if**
13: **end for**

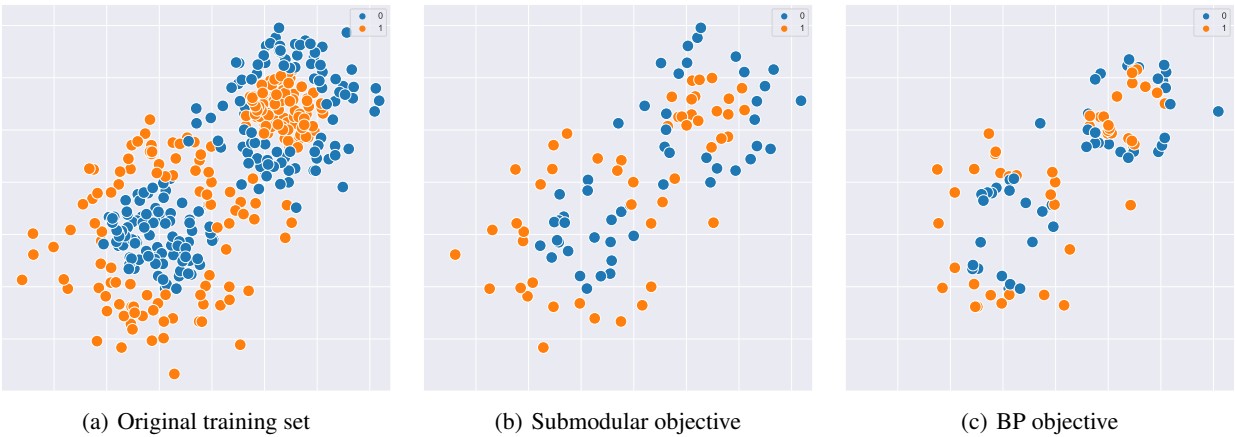

| (a) Original training set | (b) Submodular objective | (c) BP objective |

Figure 2: Greedy algorithm selection on submodular (second panel) and BP (third panel) objectives for subset selection of $100$ points of training data from a ground set of $400$ points. The first panel depicts the entire training (ground) set. The details are provided in Section I.

than "online" since it is consistent with a number of **streaming** submodular maximization algorithms [Badanidiyuru et al., 2014, Chekuri et al., 2015, Feldman et al., 2018]. The approach in [Liu et al., 2021] assume that the function $h$ is known, with arbitrary queries available, and there is no cost for evaluating it with different sets as input. However, the items are revealed one by one, using a fixed order, and the algorithm must decide whether to add the item to the set or to forever forget it. Hence, there is no statistical estimation component to their setting, and they provide competitive ratio bounds rather than regret bounds as in our present work.

# D  APPLICATIONS AND ROLE OF MNN FUNCTIONS

In this section, we present two examples that illustrate the modeling power of BP functions for different applications and compare this approach with common approaches from the literature.

## D.1  ACTIVE LEARNING

From Figure 2, we see that the BP function (third panel) results in the selection of complementary points near the decision boundary—i.e., points of opposite class that are proximal. It is *impossible* to choose a submodular function that encourages this type of desirable cooperative behavior due to the diminishing-returns property.

**Comparison with approaches for Pool-based Active Learning**  In their survey paper, Settles [2009] compare submodularity-based approaches with other approaches for active learning. The main benefit of framing the active learning problem as submodular is that the greedy algorithm can be employed, which is much less computationally expensive than other common active learning approaches. While submodularity has been shown to be relevant to active learning [Guestrin et al., 2005, Wei et al., 2015, Hoi et al., 2006], Settles [2009] remark that in general, the active learning problem cannot be framed as submodular.

In our paper, by extending the classes of functions that can be optimized online, we take a step towards addressing this limitation of submodularity. Further, an open question outlined in Settles [2009] is that of multi-task active learning, which has not been explored extensively in previous work. However, our formulation in Vignette 2 naturally extends to this multi-task setting.

## D.2  RECOMMENDATION SYSTEMS

In Table 3, the BP function enables the desirable selection of movies from the same series in the correct order. As above, it is impossible to design a submodular utility function that encourages this type of behavior.

|   | SM Objective | BP Objective |
|---|---|---|
| 0 | Lion King, The | Godfather, The |
| 1 | Speed | Godfather: Part II, The |
| 2 | Godfather, The | Godfather: Part III, The |
| 3 | Godfather: Part II, The | Star Wars: Episode I |
| 4 | Terminator, The | Memento |
| 5 | Good Will Hunting | Harry Potter I |
| 6 | Memento | Star Wars: Episode II |
| 7 | Harry Potter I | Harry Potter: II |
| 8 | Dark Knight, The | Star Wars: Episode III |
| 9 | Inception | Dark Knight, The |

Table 3: Comparison of the selections of the greedy algorithm on submodular and BP objectives for movie recommendation, on a toy ground set of 23 movies from the MovieLens dataset. The submodular objective is the facility location objective, chosen from Chen et al. [2017]. In the BP objective, there is an additional reward at each step for choosing a movie that is complementary with previously selected movies; this results the desirable joint **selection of groups of movies from the same series**. The task is formalized mathematically in Section 6, and experimental details are provided in the supplement.

**Comparison with approaches for online recommendation**  For recommender systems, the dependencies between past and future recommendations may be modeled through a changing "state variable," leading to adopting reinforcement learning (RL) solutions [Afsar et al., 2021]; these can be framed as multi-task RL problems [Bose et al., 2023, 2024]. These have been tremendously effective at maximizing engagement; however, Kleinberg et al. [2022] highlight that a key oversight of these approaches is that the click and scroll-time data that platforms observe is not representative of the users' actual utilities: "research has demonstrated that we often make choices in the moment that are inconsistent with what we actually want." Hence, Kleinberg et al. advocate to encode diminishing returns of addictive but superficial content into the model, in the manner that we do with submodular functions. Further, RL systems are incentivized to manipulate users' behavior [Wilhelm et al., 2018, Hohnhold et al., 2015], mood [Kramer et al., 2014] and preferences [Epstein and Robertson, 2015]; this inspires the use of principled mathematical techniques, as in the present work, to design systems to behave as we want rather than simply following the trail of the unreliable observed data.

| | |
|---|---|
| Lion King, The | Good Will Hunting |
| Speed | Godfather: Part III, The |
| True Lies | Star Wars: Episode I - The Phantom Menace |
| Aladdin | Gladiator |
| Dances with Wolves | Memento |
| Batman | Shrek |
| Godfather, The | Harry Potter I: The Sorcerer's Stone |
| Godfather: Part II, The | Star Wars: Episode II - Attack of the Clones |
| Terminator, The | Harry Potter II: The Chamber of Secrets |
| Indiana Jones and the Last Crusade | Star Wars: Episode III - Revenge of the Sith |
| Men in Black | Dark Knight, The |
| | Inception |

Table 4: Ground set for Table 3

# E  A SIMPLE APPROACH TO GUARANTEE LOW REGRET: WHY IT IS TOO WEAK

In this section, we provide an alternate proof for the approximation ratio that the greedy algorithm obtains on a BP function in the offline setting. The robustness of this proof can be very simply studied, in a manner similar to Chen et al. [2017]. However, the approximation ratio obtained is worse than that of Bai and Bilmes [2018]; hence, the regret guarantee in the online setting would be provided against a weak baseline. This motivates why we revisit the proof from Bai and Bilmes [2018]. Just for this section, we use simpler notation $h$ for the BP function and $k$ for the cardinality constraint, since we are presenting the argument for the offline setting.

**Proposition 1.** *For a BP maximization problem subject to a cardinality constraint,* $\max_{S:|S|\leq k} h(S)$ *where* $h(S) = f(S) + g(S)$*, the greedy algorithm obtains the following guarantee:*
$$h(S) \geq (1 - e^{-(1-\kappa^g)})h(S^*),$$
*where* $S^* = \{v_1^*, \ldots, v_k^*\} = arg\max_{S:|S|\leq k} h(S)$ *and* $\kappa^g$ *is the curvature of the supermodular function g.*

*Proof.* For $t < k$, let $S_t = \{v_1, \ldots, v_t\}$ be the items chosen by the greedy algorithm. We can write:
$$h(S^*) \leq h(S^* \cup S_t)$$
$$= h(S_t) + \sum_{j=1}^{k} h(v_j^*|S_t \cup \{v_1^*, \ldots, v_{j-1}^*\})$$
$$\leq h(S_t) + \frac{1}{1 - \kappa^g} \sum_{j=1}^{k} h(v_j^*|S_t)$$
$$\leq h(S_t) + \frac{1}{1 - \kappa^g} \sum_{j=1}^{k} h(v_{t+1}|S_t)$$
$$= h(S_t) + \frac{k}{1 - \kappa^g} \big(h(S_{t+1}) - h(S_t)\big),$$

where the first inequality uses Lemma C.1.(ii) of Bai and Bilmes [2018] and the second inequality is due to the update rule of the greedy algorithm. Rearranging the terms, we can write:
$$h(S^*) - h(S_t) \leq \frac{k}{1 - \kappa^g} \big([h(S^*) - h(S_t)] - [h(S^*) - h(S_{t+1})]\big)$$
$$h(S^*) - h(S_{t+1}) \leq (1 - \frac{1 - \kappa^g}{k})(h(S^*) - h(S_t))$$

Applying the above inequality recursively for $t = 0, \ldots, k - 1$, we have:
$$h(S^*) - h(S) \leq (1 - \frac{1 - \kappa^g}{k})^k (h(S^*) - \underbrace{h(\emptyset)}_{=0})$$

Using the inequality $1 - x \leq e^{-x}$ and rearranging the terms, we have:
$$h(S) \geq (1 - e^{-(1-\kappa^g)})h(S^*)$$

If $\kappa_f = 1$, this approximation ratio matches the obtained approximation ratio for the greedy algorithm in Theorem 3.7 of Bai and Bilmes [2018] without the need to change the original proof of the greedy algorithm.  □

# F  PROOFS FROM SECTION 3

## F.1  APPROXIMATE GREEDY ON BP FUNCTIONS

**Notation**  We use $S_t$ to refer to the ordered set of elements chosen for function $h$ until round $t$, and $S$ to refer to the ordered final set of items chosen for function $h$ until round $T$. Hence, $S_j$ refers to the first $j$ elements chosen for $h$. Let $s_j$ be the $j^{\text{th}}$ element of $S$. Then, we define $a_j = h(s_j|\{s_1 \ldots s_{j-1}\})$ be the gain of the $j^{\text{th}}$ element chosen.

Recall that $S$ is an ordered set. We let $C \subseteq [k]$ denote the indices (in increasing order) of elements in $S$ that are also in $S^*$. For instance, for $S = \{s_1 \ldots s_5\}$ and $S \cap S^* = \{s_1, s_2, s_3\}$, we have $C = \{1, 2, 3\}$. Hence, $j \in C \iff s_j \in S \cap S^*$. Further, define filtered sets $C_t = \{c \in C | c \leq t\}$ as the subset of the first $t_{th}$ elements of $S$ that are also in the optimal $S^*$.

We restate the lemma for ease of reference.

**Lemma 1.** *Any output $S$ of the approximate greedy selection rule in Equation* (2) *admits the following guarantee for BP objectives (Def.* 2*) for Problem* (1)*:*

$$h(S) \geq \frac{1}{\kappa_f} \left[ 1 - e^{-(1-\kappa^g)\kappa_f} \right] h(S^*) - \sum_{j=1}^{k} r_j,$$

*where $\kappa_f, \kappa^g$ are as defined in Definitions* 3 *and* 4*.*

*Proof of Lemma* 1*.* From Lemma 4, we have that the approximate greedy procedure obeys $k$ different inequalities, and we wish to show that this is sufficient to obey the inequality above. In order to complete the argument, we consider the worst-case overall gain if these $k$ inequalities are satisfied; and show that this worst-case sequence satisfies the desired, and hence the approximate greedy procedure must satisfy the desired as well.

To characterize the worst-case gains, we define a set of linear programming problems parameterized by a set $B$ and constants $(\xi, \rho)$.

$$T(B, \xi, \rho) = \min_b \sum_{j=1}^{k} b_j$$

$$\text{s.t.} \quad h(S^*) \leq \xi \sum_{j \in [t-1] \setminus B_{t-1}} b_j + \sum_{j \in B_{t-1}} b_j + \frac{k - |B_{t-1}|}{1 - \beta} b_t \quad , \forall t \in [k] \tag{13}$$

In the above, the decision variable $b = [b_1 \ldots b_k]$ is a vector in $\mathbb{R}^k$, and satisfies $b \geq 0$. The constants $k$ is a fixed value for the LP. The parameter of the LP, $B \subseteq [k]$, and $B_t = \{j \in B | j \leq t\}$ is the filtered set. Note that the constraints are linear in $b$ with non-negative coefficients.

The above LP becomes helpful to our setting when we set $(\xi, \beta) = (\kappa_f, \kappa^g)$. Additionally, we are interested in the choices $B = C$ and $B = \emptyset$, where $C$ is defined prior to the lemma statement. To show the result, we hope to show the following chain of inequalities:

$$h(S) + \sum_{j=1}^{k} r_j \geq T(C, \kappa_f, \kappa^g) \geq T(\emptyset, \kappa_f, \kappa^g) \geq \omega h(S^*) \tag{14}$$

In the above,

$$\omega = \frac{1}{\kappa_f} \left[ 1 - e^{-(1-\kappa^g)\kappa_f} \right]$$

Combining the two ends of this chain yields the desired lemma statement. We recognize that $T(\cdot)$ is exactly the LP considered in Bai and Bilmes [2018], modulo notation differences. Since the second and third inequality are just statements about the linear program, they follow directly from Lemma D.2 in Bai and Bilmes [2018] when we substitute $\xi = \kappa_f$ and $\beta = \kappa^g$.

For the first inequality, we have from Lemma 4 that $b_j = a_j + r_j$ is a feasible solution for the linear program $T(C, \kappa_f, \kappa^g)$. Hence,

$$T(C, \kappa_f, \kappa^g) \leq \sum_{j=1}^{k} b_j = \sum_{j=1}^{k} a_j + \sum_{j=1}^{k} r_j = h(S) + \sum_{j=1}^{k} r_j$$

$\square$

The lemma below is a modified version of Equation (19) in Bai and Bilmes [2018], which accounts for the deviation of our algorithm from the greedy policy.

**Lemma 4.** *Using the notation above and for $S$ as chosen by the approximate greedy procedure, it follows that $\forall t \in [k]$,*

$$h(S^*) \leq \kappa_f \sum_{j \in [t-1] \setminus C_{t-1}} (a_j + r_j) + \sum_{j \in C_{t-1}} (a_j + r_j) + \frac{k - |C_{t-1}|}{1 - \kappa^g} (a_t + r_t)$$

*Proof of Lemma 4.* By the properties of BP functions from Lemma C.2 in Bai and Bilmes [2018], it follows for all $t \in [k]$

that

$$h(S^*) \leq \kappa_f \sum_{j \in [t-1] \setminus C} a_j + \sum_{j \in C_{t-1}} a_j + h(S^* \setminus S_{t-1} | S_{t-1}) \tag{15}$$

$$\leq \kappa_f \sum_{j \in [t-1] \setminus C} (a_j + r_j) + \sum_{j \in C} (a_j + r_j) + h(S^* \setminus S_{t-1} | S_{t-1}) \tag{16}$$

The inequality above follows because the coefficients on the first two summations are positive and $r_j \geq 0$. Now, we must simplify the third term to obtain the desired. For any feasible $v$,

$$h(v | S_{t-1}) \leq \sup_v h(v | S_{t-1}) \leq h(s_t | S_{t-1}) + r_t. \tag{17}$$

The first inequality follows from the definition of $\sup$ and the second follows from the definition of $r_t$ in the proof of Theorem 1 above. Now, apply inequality (iv) from Lemma C.1 in Bai and Bilmes [2018]:

$$h(S^* \setminus S_{t-1} | S_{t-1}) \leq \frac{1}{1 - \kappa^g} \sum_{v \in S^* \setminus S_{t-1}} h(v | S_{t-1})$$

$$\leq \frac{1}{1 - \kappa^g} \sum_{v \in S^* \setminus S_{t-1}} h(s_t | S_{t-1}) + r_t$$

The second line follows from Equation (17).

We have that

$$|S^* \setminus S_{t-1}| = |S^*| - |S^* \cap S_{t-1}| = k - |S^* \cap S_{t-1}|.$$

Hence,

$$h(S^* \setminus S_{t-1} | S_{t-1}) \leq \frac{k - |S^* \cap S_{t-1}|}{1 - \kappa^g} [h(s_t | S_{t-1}) + r_t]$$

Recognizing that $|S^* \cap S_{t-1}| = |C_{t-1}|$ completes the argument. $\qquad \square$

## F.2 APPROXIMATE GREEDY ON WS FUNCTIONS

Define $S, s_j, a_j, C$ as in the proof for BP functions. Below, we present the counterparts of the lemmas in the proof of the BP functions for the present case. The proof for Lemma 5 is different than Lemma 4 due to the change in the class of functions being considered. The similarity of the two proofs suggests the generality of our proof technique and indicates that it may be analogously applied to other classes of functions as well. We restate the Lemma for ease of reference.

**Lemma 2.** *Any output $S$ of the approximate greedy selection rule in Equation* (2) *admits the following guarantee on objectives with submodularity ratio $\gamma$ and generalized curvature $\zeta$ (Definitions 5 and 6) for Problem* (1):

$$h(S) \geq \frac{1}{\zeta} \left( 1 - e^{-\zeta \gamma} \right) h(S^*) - \sum_{j=1}^{k} r_j.$$

*Proof of Lemma 2.* We consider again the parameterized LP $T(\cdot)$, but this time with the constants set as $\xi = \zeta, \rho = 1 - \gamma$. To show the result, we hope to show the following chain of inequalities:

$$h(S) + \sum_{j=1}^{k} r_j \geq T(C, \zeta, 1 - \gamma) \geq T(\phi, \zeta, 1 - \gamma) \geq \omega h(S^*) \tag{18}$$

In the above,

$$\omega = \frac{1}{\zeta} \left[ 1 - \left( 1 - \frac{\gamma \zeta}{k} \right)^k \right]$$

Similarly to the argument in Lemma 1, the first two inequalities follow directly from Lemma D.2 in Bai and Bilmes [2018] when we substitute $\xi = \zeta$ and $\rho = 1 - \gamma$. Under the same choice of constants, we have from Lemma 5 that $b_j = a_j + r_j$ is a feasible solution for the linear program $T(C, \zeta, 1 - \gamma)$. Hence,

$$T(C, \zeta, 1 - \gamma) \leq \sum_{j=1}^{k} b_j = \sum_{j=1}^{k} a_j + \sum_{j=1}^{k} r_j = h(S) + \sum_{j=1}^{k} r_j$$

Recognizing that

$$\frac{1}{\zeta}\left[1 - \left(1 - \frac{\gamma\zeta}{k}\right)^k\right] \geq 1 - e^{-\zeta\gamma}$$

completes the argument. □

The lemma below is a modified version of Lemma 1 in Bian et al. [2019].

**Lemma 5.** *Using the notation above and for $S$ as chosen by the approximate greedy procedure, it follows that $\forall t \in \{0\ldots k-1\}$,*

$$h(S^*) \leq \zeta \sum_{j\in[t]\backslash C_t}(a_j + r_j) + \sum_{j\in C_t}(a_j + r_j) + \frac{1}{\gamma}(k - |C_t|)(a_{t+1} + r_{t+1})$$

*Proof of Lemma 5.* The proof follows from the definitions of generalized curvature, submodularity ratio, and instantaneous regret $r_t$.

$$h(S^* \cup S_t) = h(S^*) + \sum_{j\in[t]} h(s_j|S^* \cup S_{j-1})$$

We can split the summation above to separately consider the elements from $S_t$ that do and do not overlap with $S^*$.

$$h(S^* \cup S_t) = h(S^*) + \sum_{j:s_j\in S_t\backslash S^*} h(s_j|S^* \cup S_{j-1}) + \underbrace{\sum_{j:s_j\in S_t\cap S^*} h(s_j|S^* \cup S_{j-1})}_{=0}$$

$$= h(S^*) + \sum_{j:s_j\in S_t\backslash S^*} h(s_j|S^* \cup S_{j-1}) \tag{19}$$

From the definition of submodularity ratio,

$$h(S^* \cup S_t) \leq h(S^*) + \frac{1}{\gamma}\sum_{\omega\in S^*\backslash S_t} h(\omega|S_t) \tag{20}$$

From the definition of generalized curvature, it follows that

$$\sum_{j:s_j\in S_t\backslash S^*} h(s_j|S^* \cup S_{j-1}) \geq (1-\zeta)\sum_{j:s_j\in S_t\backslash S^*} h(s_j|S_{j-1})$$

$$= (1-\zeta)\sum_{j:s_j\in S_t\backslash S^*} a_{j+1} \tag{21}$$

Then, plugging the inequalities (20) and (21) into (19),

$$h(S^*) = h(S^* \cup S_t) - \sum_{j:s_j\in S_t\backslash S^*} h(s_j|S^* \cup S_{j-1})$$

$$\leq \left[h(S) + \frac{1}{\gamma}\sum_{\omega\in S^*\backslash S} h(\omega|S)\right] + \left[\zeta\sum_{j:s_j\in S_t\backslash S^*} a_{j+1} - \sum_{j:s_j\in S_t\backslash S^*} a_{j+1}\right] \tag{22}$$

Now, we can rearrange and write

$$h(S) - \sum_{j:s_j\in S_t\backslash S^*} a_{j+1} = \sum_{j:s_j\in S_t\cap S^*} a_{j+1}$$

to simplify Equation (22) as

$$h(S^*) = \zeta \sum_{j:s_j \in S_t \setminus S^*} a_{j+1} + \frac{1}{\gamma} \sum_{\omega \in S^* \setminus S} h(\omega|S) + \sum_{j:s_j \in S_t \cap S^*} a_{j+1}$$

$$\leq \zeta \sum_{j:s_j \in S_t \setminus S^*} a_{j+1} + \frac{1}{\gamma} \sum_{\omega \in S^* \setminus S} (a_{t+1} + r_t) + \sum_{j:s_j \in S_t \cap S^*} a_{j+1} \tag{23}$$

$$\leq \zeta \sum_{j:s_j \in S_t \setminus S^*} (a_{j+1} + r_j) + \sum_{j:s_j \in S_t \cap S^*} (a_{j+1} + r_j) + \frac{1}{\gamma}(k - |C_t|)(a_{t+1} + r_t) \tag{24}$$

Equation (23) follows by using the definitions of $r_t$ and supremum, and Equation (24) follows since $r_t \geq 0$. □

## F.3 APPROXIMATE WEIGHTED GREEDY ON BP FUNCTIONS

We recap notation for ease of reference. Define the modular lower bound of the submodular function $l_1(S) = \sum_{j \in S} f(j|V \setminus \{j\})$. Additionally, define the totally normalized submodular function as $f_1(S) = f(S) - l_1(S)$. Note that the $f_1$ will always have curvature $\kappa_f = 1$. $h(S) = f_1(S) + g(S) + l_1(S)$. Now, define the function

$$\pi_j(v|A) = \left(1 - \frac{1}{k}\right)^{k-j-1} f_1(v|A) + g(v|A) + l_1(v) \tag{25}$$

Also define

$$\pi_j(A) = \left(1 - \frac{1}{k}\right)^{k-j} f_1(A) + g(A) + l_1(A) \tag{26}$$

The proof of Lemma 7 follows using the same outline as the approximation-guarantee of the distorted greedy algorithm for BP functions (Theorem 3 in Liu et al. [2022]). However, Algorithm 4 is not greedy, so we keep record of the deviation of Algorithm 4 from the best-in-hindsight choice at *each stage*. This results in the new second term in Lemma 3.

**Lemma 3.** *Any output $S$ of the approximate distorted greedy selection rule in Equation* (4) *admits the following guarantee for Problem* (1) *with a BP objective (Def. 2):*

$$h(S) \geq \min \left\{1 - \frac{\kappa_f}{e}, \ 1 - \kappa^g\right\} h(S^*) - \sum_{j=1}^{k} r_j,$$

*where $\kappa_f, \kappa^g$ are as defined in Def. 3 and 4.*

*Proof.* Using the submodular and supermodular curvature definition, we can write:

$$l_1(S) = \sum_{j \in S} f(j|V \setminus \{j\}) \geq (1 - \kappa_f)f(S)$$

$$l_2(S) = \sum_{j \in S} g(j|\emptyset) \geq (1 - \kappa^g)g(S)$$

Define $l = l_1 + l_2$. Then, we can use the result of Lemma 7 to write:

$$f(S) + g(S) = f_1(S) + g_1(S) + l(S)$$

$$\geq \left(1 - \frac{1}{e}\right) f_1(S^*) + l(S^*) - \sum_{j=1}^{k} r_j$$

$$= \left(1 - \frac{1}{e}\right) (f(S^*) - l_1(S^*)) + l_1(S^*) + l_2(S^*) - \sum_{j=1}^{k} r_j$$

$$= \left(1 - \frac{1}{e}\right) f(S^*) + \frac{1}{e} l_1(S^*) + l_2(S^*) - \sum_{j=1}^{k} r_j$$

$$\geq \left(1 - \frac{1}{e}\right) f(S^*) + \frac{1 - \kappa_f}{e} f(S^*) + (1 - \kappa^g) g(S^*) - \sum_{j=1}^{k} r_j$$

$$= \left(1 - \frac{\kappa_f}{e}\right) f(S^*) + (1 - \kappa^g) g(S^*) - \sum_{j=1}^{k} r_j$$

$$\geq \min\left\{1 - \frac{\kappa_f}{e},\ 1 - \kappa^g\right\} h_q(S^*) - \sum_{j=1}^{k} r_j.$$

$\square$

**Lemma 6.**

$$\pi_j(s_j | S_{j-1}) + r_j \geq \frac{1}{k}\left(1 - \frac{1}{k}\right)^{k-(j+1)} (f(S^*) - f(S_j) + \frac{1}{k} l(S^*)$$

*Proof of 6.* From the definition of $r_j$,

$$\pi_j(s_j | S_{j-1}) + r_j \geq \frac{1}{k} \sum_{e \in S^*} \pi_j(e | S_{j-1})$$

$$= \frac{1}{k} \sum_{e \in S^*} \left(1 - \frac{1}{k}\right)^{k-(j+1)} f_1(e | S_{j-1}) + g_1(e | S_{j-1}) + l(e)$$

$$\geq \frac{1}{k}\left(1 - \frac{1}{k}\right)^{k-(j+1)} (f_1(S^*) - f_1(S_{j-1})) + \frac{1}{k} l(S^*)$$

The inequality follows from the submodularity of $f_1$ and the supermodular curvature of $g_1$.

$\square$

**Lemma 7.** *Any approximately weighted greedy procedure with constants $\{r_j\}_{j=1}^{k}$ returns a set $S$ of size $k$ such that*

$$f_1(S) + g_1(S) + l(S) + \sum_{j=1}^{k} r_j \geq \left(1 - \frac{1}{e}\right) f_1(S^*) + l(S^*)$$

*Proof.* According to the definition of $\pi$, we have that $\pi_0(\emptyset) = 0$ and

$$\pi_k(S) = f_1(S) + g_1(S) + l(S)$$

Applying Lemma 4 from Liu et al. [2022], we have

$$\pi_{j+1}(S_{j+1}) - \pi_j(S_j)$$

$$= \pi_j(s_{j+1}|S_j) + \frac{1}{k}\left(1 - \frac{1}{k}\right)^{k-(j+1)} f_1(S_j)$$

$$\geq \frac{1}{k}\left(1 - \frac{1}{k}\right)^{k-(j+1)} f_1(S^*) + \frac{1}{k}l(S^*) - r_{j+1}$$

Above, we applied Lemma 6 to obtain the inequality. Now, we have that

$$f_1(S) + g_1(S) + l(S) = \sum_{j=0}^{k-1} \pi_{j+1}(S_{j+1}) - \pi_j(S_j)$$

$$\geq \sum_{j=0}^{k-1} \frac{1}{k}\left(1 - \frac{1}{k}\right)^{k-(j+1)} f_1(S^*) + \frac{1}{k}l(S^*) - r_{j+1}$$

$$\geq \left(1 - \frac{1}{e}\right) f_1(S^*) + l(S^*) - \sum_{j=1}^{k} r_j$$

$\square$

# G   DISCUSSION AND PROOFS FROM SECTION 4

## G.1   REMARKS ON HYPERPARAMETERS $(\eta, b)$

Note that $b$ refers to our budget fraction variable as it serves to limit the final size of $G_t$, while $\eta$ is an accuracy-computation tradeoff variable that tends to produce larger $G_t$'s. While $\eta$ and $b$ are somewhat related (and are partially redundant) we utilize the "budget" and "accuracy" notion as originally defined in Zenati et al. [2022] to be consistent with that work

## G.2   REMARKS ON STEP SIZE $\beta_t$

From on the analysis found in Zenati et al. [2022], we set

$$\beta_t = \sqrt{\lambda}B + \sqrt{4\log(T) + \log\left(e + \frac{et}{\lambda}\right)d_{\text{eff}}} \tag{27}$$

which enables our regret bounds to hold where $e = \exp(1)$, $\lambda$ is a hyperparameter, and $B$ is our RKHS norm bound.

In our empirical simulations, however, we found it much more effective to set $\beta_t$ to a constant which is then tuned as a hyperparameter. In fact, Zenati et al. [2022] found this to be the case in their simulations as well.

## G.3   REMARK ON ROLE OF KERNEL PARAMETERS ON $d_{\text{eff}}$

Consider the RBF kernel $k(x, x') = \exp(-b\|x - x'\|^2)$. If the parameter $b$ is very large, then the kernel function will be very close to zero for all $x \neq x'$. Hence, the kernel matrix $K_T$ will be close to the identity matrix, and the eigenvalues will decay very slowly. Hence the effective dimension $d_{\text{eff}}$ is likely to be large. Our current regret bound does not capture this, because we wanted to focus on the scaling of regret with $T$. However, there is a constant in front that scales as $b$, which effectively changes the base of the $\log(T)$ in the regret bound [Seeger et al., 2008, Section 4.B]. In Figure 3, we see that if the horizon $T$ is quite small, this effect can dominate and make the $T$-scaling appear almost linear. On the other hand, if we make $b$ very small, then the quantity $B$ would increase; this is because $k(x, x')$ being large is not very informative about the function values at $x$ and $x'$. Hence, some care is required to tune the kernel parameters correctly. This applies to other kernel functions as well. This effect is present in prior works [Srinivas et al., 2010, Krause and Ong, 2011, Zenati et al., 2022] as well, but these do not address it explicitly which is why we wanted to offer some clarity about this point.

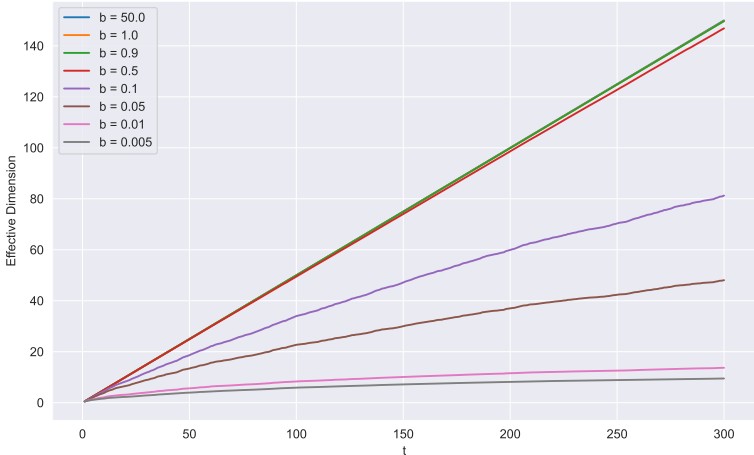

Figure 3: The dependence of effective dimension $d_{\text{eff}}$ as on the parameter $b$ in the RBF kernel.

### G.4 PROOF OF THEOREM 1(B)

The proof follows the same LP construction of Conforti and Cornu'ejols [1984] as Theorem 1. The main contribution lies showing Lemma 5; this shows that the offline counterpart, Lemma 1 from Bian et al. [2019], holds in the online setting as well.

*Proof.* Define $r_t$ and $R_t$ as in the proof of Theorem 1. From Lemma 2 applied to each $h_q$, it follows that

$$\mathcal{R}_{\text{WS}}(T) \leq \sum_{k=1}^{m} \sum_{t=1}^{T} \mathbb{I}(u_t = k) r_t = R_T$$

As in the proof of Theorem 1, combining Theorem 4.1 of Zenati et al. [2022] with the above inequality, our argument is complete. $\qquad\square$

## H DISCUSSION AND PROOFS FROM SECTION 5

### H.1 REMARKS ON Liu et al. [2022]

**Comparison of $\alpha$ for greedy vs weighted Greedy.**  In Figure 4, we compare the $\alpha$ of the greedy optimization of the BP function in Bai and Bilmes [2018] with the distorted greedy variant in Equation (12). In the left panel, we see that the $\alpha$ in Equation (12) is everywhere greater.

**Error in Liu et al. [2022] and proposed fix.**  In Liu et al. [2022] on the bottom of Pg.188, the authors use the inequality:
$$\sum_{e \in \text{OPT}} g_1(e|S_t) \geq (1 - \kappa^g)(g_1(OPT) - g_1(S_t))$$
Consider the following counterexample with $|V| = 3$ and $k = 2$ as the cardinality constraint. Define $g(S) = |S|^2$, which is a concave over modular function, so it is supermodular. We can verify from definitions that $\kappa^g = 0.8$ and the modular lower bound $l_2(S) = |S|$, so that $g_1(S) = |S|^2 - |S|$. For simplicity, consider the case where $t = 0$, so that $S_t = \emptyset$. Then, plugging into the equation, we see that the LHS is 0, whereas the RHS is $0.2 \times 4 = 0.8 > 0$. Hence, this is a contradiction. This example can be easily generalized to any concave over modular function, larger ground set sizes or different $t$.

We rectify this by swapping this inequality with
$$g_1(e|S_t) \geq (1 - \kappa^{g_1})(g_1(\text{OPT}) - g_1(S_t)) = 0$$
The equality above holds because $\kappa^{g_1} = 1$ by construction. Hence, the $g_1$ term disappears from the analysis. In the analysis below, we make this fix and propagate the consequences; the modified algorithm, analysis and result applies to the offline setting of Liu et al. [2022] as well.

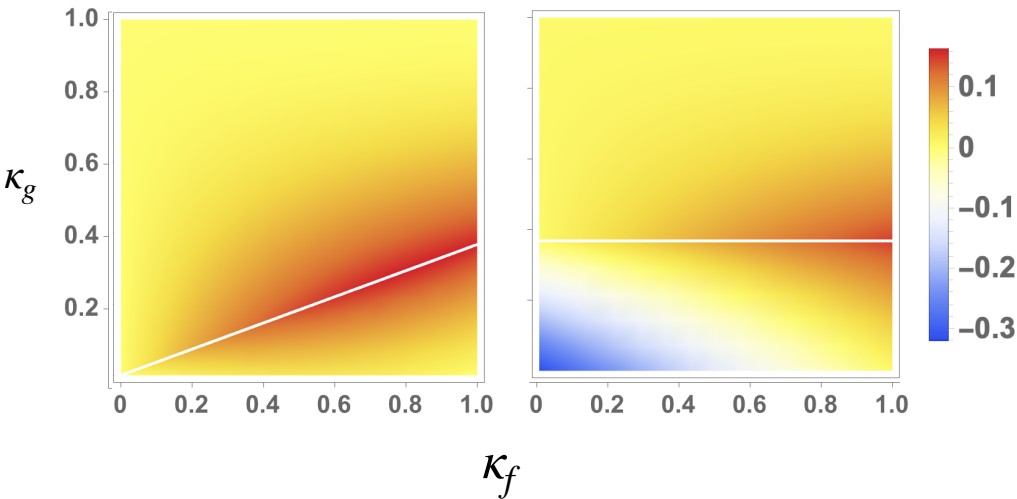

Figure 4: Contour plot of (left) $F_1(\kappa_{f,q}, \kappa_q^g) = \min\left\{1 - \frac{\kappa_{f,q}}{e}, 1 - \kappa_q^g\right\} - \frac{1}{\kappa_{q,f}}\left[1 - e^{-(1-\kappa_q^g)\kappa_{q,f}}\right]$ and (right) $F_2(\kappa_{f,q}, \kappa_q^g) = \min\left\{1 - \frac{1}{e}, 1 - \kappa_q^g\right\} - \frac{1}{\kappa_{q,f}}\left[1 - e^{-(1-\kappa_q^g)\kappa_{q,f}}\right]$. (Left) compares the $\alpha$ from Theorem 2 with that from Theorem 1, and (right) compares $\alpha$ from Proposition 2 with that from Theorem 1.

### H.2 PROOF OF SEPARATE FEEDBACK GUARANTEE

*Proof of Theorem 2.* First, we define some notation.

$$l_{q,1}(S) = \sum_{j \in S} f_q(j | V \backslash \{j\})$$

$$f_{q,1}(S) = f_q(S) - l_{q,1}(S)$$

$$l_{q,2}(S) = \sum_{j \in S} g_q(j | \emptyset)$$

$$g_{q,1}(S) = g_q(S) - l_{q,2}(S)$$

$$l_q(S) = l_{q,1}(S) + l_{q,2}(S)$$

We restrict attention to the $q$-th function $h_q$. Recall that $S_{j,q}$ refers to the first $j$ elements chosen for $h_q$.

Let the distorted objective for user $q$ when selecting the $j$-th item in the set be:

$$\pi_{j,q}(S) = \left(1 - \frac{1}{T_q}\right)^{T_q - j} f_{q,1}(S) + g_{q,1}(S) + l_q(S)$$

Additionally, define

$$\Lambda_{j,q}(x, A) = \left(1 - \frac{1}{T_q}\right)^{T_q - (j+1)} f_{q,1}(x|A) + g_{q,1}(x|A) + l_q(x)$$

As previously, we define the instantaneous regret at round $t$ as the difference between the maximum possible utility that is achievable in the round and the actual received utility. However, this time, $r_t$ is defined in terms of the distorted objective. *This is a key difference from the earlier arguments that is crucial to the current proof.*

$$r_t = \sup_{v \in V} \Lambda_{q,t_{u_t}}(v, S_{u_t, t-1}) - \Lambda_{q,t_{u_t}}(v_t, S_{u_t, t-1})$$

Define the accumulated instantaneous regret until round $t$ as

$$R_t = \sum_{j=1}^{t} r_j$$

Recognize that $R_t$ is different than $\mathcal{R}_t$. From Lemma 3 applied to each $h_q$, it follows that

$$\mathcal{R}_T \leq \sum_{q=1}^{m} \sum_{t=1}^{T} \mathbb{I}(u_t = q) r_t = R_T \tag{28}$$

Now, we can model the problem of the present work as a contextual bandit problem in the vein of Zenati et al. [2022]. Here, the context in the $t$-th round is $z_t = (\phi_{u_t}, S_{t_{u_t}, u_t})$. Now we invoke Theorem 4.1 in Zenati et al. [2022], Thus, we have that

$$\mathbb{E}[R_T] \leq O\left(\sqrt{T}\left(B\sqrt{\lambda d_{\text{eff}}} + d_{\text{eff}}\right)\right)$$

Combining this with Inequality (28), our argument is complete.

$\square$

## H.3 ANALYSIS WITHOUT Assumption (3a)

In certain applications, Assumption (3a) on $l_{q,1}$ may not be reasonable. For these cases, we may modify the algorithm slightly, and provide an alternative bound, that is slightly weaker. Consider a modified version of Algorithm 4, where line 3 is substituted with:

$$\text{Set } y_t = (1 - 1/T_{u_t})^{T_{u_t} - (t_{u_t} + 1)} y_{f,t} + y_{g,t} \tag{29}$$

Recognize that this Algorithm does not require Assumption (3a).

Define

$$\mathcal{R}_{\text{BP},3}(T) := \sum_{q=1}^{m} \min\left\{1 - \frac{1}{e}, \ 1 - \kappa_q^g\right\} h_q(S_q^*) - h_q(S_q). \tag{30}$$

Observe from the right panel of Figure 4 that the $\alpha$ in the definition above is still better than that of Bai and Bilmes [2018] for most choices of $\kappa_{f,q}, \kappa_q^g$. Now, we can state our modified result. The proof follows similarly to Theorem 2.

**Proposition 2.** *Let Assumption 1 and 2 and Assumption (3b) hold. Additionally, let the conditions on $\epsilon_t$ hold as in Theorem 2. Then Algorithm 4 with the modification above yields $\mathbb{E}[\mathcal{R}_{BP,3}(T)] \leq O\left(\sqrt{T}\left(B\sqrt{\lambda d_{eff}} + d_{eff}\right)\right)$*

*Proof of Proposition 2.* For the modified version of the algorithm described in equation (29), the analysis is almost identical. Repeating the analysis of Lemma 7 and Lemma 6, we obtain:

$$f_q(S_q) + g_{q,1}(S_q) + l_{q,2}(S_q) + \sum_{j=1}^{T_q} r_{m_j} \geq \left(1 - \frac{1}{e}\right) f_q(S_q^*) + l_{q,2}(S_q^*)$$

Then, we can follow the same arguments as in Lemma 3 to conclude:

$$f_q(S) + g_q(S) \geq \min\left\{1 - \frac{1}{e}, \ 1 - \kappa_q^g\right\} h_q(S_q^*) - \sum_{j=1}^{T_q} r_{m_j}.$$

$\square$

## H.4 REMARKS ON GUESS-AND-DOUBLE TECHNIQUE TO REPLACE Assumption (3b)

In this section, we provide a heuristic argument for why we expect that guess-and-double techniques should not affect the overall regret scaling in Theorem 2.

In the traditional multi-armed bandit, when the time horizon $T$ is unknown, the proposed method of dealing with this is to start with an initial guess $\widehat{T} = 1$ and then double each time the current time step crosses our latest guess. Any parameters in the algorithm that depend on $T$ (step size for e.g) are set based on $\widehat{T}$ instead. This divides the entire horizon into phases, one for each guess $\widehat{T}$. Then, for each phase, the regret must be sublinear because this is equivalent to playing a shorter game with known horizon. Since the regret is the accumulation of the regrets of each phase, the overall regret must be sublinear as well.

However, in our case, the situation is more intricate because the overall regret is not expressible as the summation of regret over phases. Hence, the original style of argument does not apply. What we do then, is to keep track of the change in regret due to setting the distortion co-efficient in terms of $\widehat{T_q}$ instead of $T_q$. We choose $\widehat{T_q} = \min\{2^j : j > t\}$.

When $T_q$ is known, the distortion $D_t = \left(1 - \frac{1}{T_{u_t}}\right)^{T_{u_t} - t_{u_t} - 1}$ increases monotonically from $\left(1 - \frac{1}{T_{u_t}}\right)^{T_{u_t} - 1}$ to 1 with $t_{u_t}$ i.e as more elements are added. This monotonicity is used in the original argument to obtain the sublinear regret guarantee.

However, when the guess-and-double technique is used, the distortion is no longer monotonic in $t_{u_t}$. Within each phase, $D_t$ increases from $\left(1 - \frac{1}{\widehat{T_{u_t}}}\right)^{\widehat{T_{u_t}}-1}$ to 1 but then reduces once $\widehat{T_{u_t}}$ is updated at the end of the phase. It turns out that the regret actually decreases within the phase (compared to the situation where we know $T_q$) due to the increased distortion, but increases in the transitions between the phases. Below, we characterize the changes in regret in the two cases.

Define

$$\widehat{\Lambda}_{j,q}(x, A) = \left(1 - \frac{1}{\widehat{T_q}}\right)^{\widehat{T_q}-(j+1)} f_{q,1}(x|A) + g_{q,1}(x|A) + l_q(x)$$

Analogously, we can define

$$\widehat{\pi}_{j,q}(S) = \left(1 - \frac{1}{\widehat{T_q}}\right)^{\widehat{T_q}-j} f_{q,1}(S) + g_{q,1}(S) + l_q(S)$$

**Case 1: Within phase** Previously Lemma 4 from Liu et al. [2022], we had

$$\pi_{j+1,q}(S_{j+1,q}) - \pi_{j,q}(S_{j,q})$$
$$= \Lambda_{j,q}(s_j, S_{j,q}) + \frac{1}{T_q}\left(1 - \frac{1}{T_q}\right)^{T_q-(j+1)} f_{q,1}(S_{j,q})$$

Now, we can replace this conclusion with

$$\widehat{\pi}_{j+1,q}(S_{j+1,q}) - \widehat{\pi}_{j,q}(S_{j,q})$$
$$= \widehat{\Lambda}_{j,q}(s_j, S_{j,q}) + \frac{1}{\widehat{T_q}}\left(1 - \frac{1}{\widehat{T_q}}\right)^{\widehat{T_q}-(j+1)} f_{q,1}(S_{j,q})$$
$$= \widehat{\Lambda}_{j,q}(s_j, S_{j,q}) + \frac{1}{\widehat{T_q}}\left(1 - \frac{1}{\widehat{T_q}}\right)^{\widehat{T_q}-(j+1)} f_{q,1}(S_{j,q}) + \underbrace{\left(\frac{1}{\widehat{T_q}} - \frac{1}{T_q}\right)\left(1 - \frac{1}{\widehat{T_q}}\right)^{\widehat{T_q}-(j+1)} f_1(S_{j,q})}_{N_{\text{within},j}}$$

The term $N_{\text{within},j}$ is a new term. The remainder of the proof goes through as expected, while these additional terms propagate through the proof.

**Case 2: Between phase** Note that if step $j$ is in a different phase than step $j+1$, it follows that the distortion at step $j$ is

$$\left(1 - \frac{1}{\widehat{T_q}}\right)^{\widehat{T_q}-\widehat{T_q}} = 1.$$

Since step $t+1$ is the first time step in a phase, it follows that the guess for $\widehat{T_q}$ just doubled, and is $\widehat{T_q} = 2i$. Then, the distortion for step $j+1$ is

$$\left(1 - \frac{1}{2i}\right)^{t-1}$$

As in the Case 1, we can track the extra term from Lemma 4, which in this case is

$$N_{\text{between},j} = -\left(1 - \left(1 - \frac{1}{2i}\right)^{t-1}\right) f_1(S_{j,q})$$

As before, this new term propagates through the proof.

**Putting it together** Accounting for the new terms, our modified final statement of Lemma 3

$$h_q(S_q) \geq \min\left\{1 - \frac{\kappa_{f,q}}{e}, \ 1 - \kappa_q^g\right\} h_q(S_q^*) - \sum_{j=1}^{T_q} r_{m_j} + \sum_{j:\text{change}} N_{\text{between},j} + \sum_{j:\text{no change}} N_{\text{within},j}$$

Above the indices $(j : \text{change})$ include the $\log(T_q)$ time steps, which are the first time step in a phase i.e the first time step after our guess $\widehat{T_q}$ was recently updated; the indices $(j : \text{no change})$ include all other time steps. Hence, the new term

$$N = \sum_{j:\text{change}} N_{\text{between},j} + \sum_{j:\text{no change}} N_{\text{within},j}$$

gets subtracted from the regret. We observe that each of the $N_{\text{between},j}$ terms are positive and there are many of these: $T_q - \log(T_q)$ to be precise. However, the $N_{\text{within},j}$ terms are negative and increase the regret; however, there are only $\log(T_q)$ of these. While it is difficult to quantify the terms exactly, there is no strong reason to believe that the few negative terms greatly outweigh the positive terms. From preliminary simulations, we find that the regret remains roughly the same with the doubling trick; we leave an extensive experimental investigation of this to future work.

# I  DETAILS ON EXPERIMENTS

**Details for Table 4, Table 3**  The chosen toy ground set of 23 elements is detailed in Table 4. The submodular function is the facility location function; we chose this function because it is used in prior work Chen et al. [2017] for the task of movie recommendation. The supermodular part is the sum-sum-dispersion function, and the weights that capture the complementarity between movies are specified in the python notebook `code/table-1.ipynb` in the attached code.

From Table 3, we notice that with the submodular objective, the greedy algorithm chooses the first two movies in the Godfather series but does not choose the third. Similarly, it chooses the first Harry Potter but not the subsequent ones. In contrast, with the BP function, the greedy algorithm chooses all elements from the series in both cases. This behavior cannot be encoded using solely a submodular function, but it is very easy to do so with a BP function.

**Setup for movie recommendation in Figure 1**  From MovieLens and using the matrix-completion approach in Cai et al. [2010], we obtain a ratings matrix $M \in \mathbb{R}^{900 \times 1600}$, where $M_{i,j}$ is the rating of the $i_{th}$ user for the $j^{\text{th}}$ movie; for density of data, we consider the most active users and most popular movies.

We cluster the users into $m = 10$ groups using the $k$-means algorithm and design a BP objective for each user-group. The objective for the $q_{th}$ group is decomposed as $h_q(A) = \sum_{v \in A} m_q(v) + \lambda_1 f_q(A) + \lambda_2 g_q(A)$, where the modular part $m_q(v)$ is the average rating for movie $v$ amongst all users in group $k$.

Let the set $L$ refer to the collection of all genres in the ground set. The concave-over-modular submodular part encourages the recommender to maintain a balance across genres in chosen suggestions: $f_q(A) = \sum_{g \in L} \sqrt{1 + u_{q,g}(A)}$. The set $L$ is the collection of all genres. We now specify what $u_{q,g}(\cdot)$ is. For each element $v \in V$, define a vector $r(v) \in \{0,1\}^{|L|}$. Here, each entry corresponds to a genre and is 1 if the genre is associated with the movie $v$. Then let $N_v = r(v)^{\top} \mathbf{1}$ denote the number of genres for movie $v$. In $f_q(\cdot)$, we specify

$$u_{q,g}(A) = \sum_{v \in A} \mathbf{1}(m_q(v) > \tau) \frac{\mathbf{1}(\text{v has genre g})}{N_v}$$

Above, $\mathbf{1}$ is the indicator function.

The supermodular function, in contrast is designed to encourage the optimizer to exploit complementarities within genres $g_q(A) = \sum_{g \in L} (1 + \tilde{u}_{q,g}(A))^2$, where we define

$$\tilde{u}_{q,g}(A) = \sum_{v \in A} \mathbf{1}(\text{v has genre g}(m_k(v) > \tau)) \frac{m_q(v)}{N_v}$$

We want the complementarities to be amplified when the movies have higher ratings, so notice that each term in $\tilde{u}_{q,g}$ is scaled by $m_q(v)$ relative to each term of $u_{q,g}$. The constants $\lambda_1, \lambda_2$ were chosen such that the supermodular part slightly dominates the submodular part, since previous works already study functions that are primarily submodular. The code is contained in notebook "Figure 2."

**Kernel Estimation for Figure 1**  For Algorithm 1, we choose the RBF kernel for movies, the linear kernel for users and the Jaccard kernel for a history of recommendations. The composite kernel $\hbar((u, v, A), (u', v', A')) = \kappa_1 \hbar_{\text{user}}(u, u') + \kappa_2 \hbar_{\text{movie}}(v, v') + \kappa_3 \hbar_{\text{history}}(A, A')$ for $\kappa_1, \kappa_2, \kappa_3 > 0$. For Algorithm 4, we choose the RBF kernel for $o_t$.

**Active Learning.**  This corresponds to Vignette 2 with $m = 1$ tasks. We apply the Naive-Bayes formulation of active learning in Equation (5) of Wei et al. [2015] and set the submodular part as $f(A) = f^{\text{NB}}(A)$. The supermodular part is the sum-sum-dispersion function as above $g(A) = \sum_{v_t \in A} \sum_{v_j \in A : v_j \neq v_t} B_{t,j}$. Here $B_{t,j} = 0$ if $(v_t, v_j)$ are from the same

class, and $B_{t,j} = 1/\text{dist}(v_t, v_j)$ if $(v_t, v_j)$ are from the opposite class; this encourages the selection of proximal points from different classes.

Here, we elaborate on the choice of submodular function. Assume our features are discrete - each point $v \in V$ has features $x_v \in \mathcal{X}$ (where $\mathcal{X}$ is some finite set) and binary label $y_v \in \{0, 1\}$, denoted by the orange and blue colors in Figure 2. Then, for any $(x \in \mathcal{X}, y \in \{0, 1\})$ and for any subset of training points $S \subseteq V$, we can define

$$m_{x,y}(S) = \sum_{v \in S} \mathbf{1}(x_v = x \wedge y_v = y)$$

as the empirical count of the joint occurrence of $(x, y)$ in $S$. Then, inspired by the construction in Wei et al., we define the submodular part $f$ as

$$f(S) = \sum_{x \in \mathcal{X}} \sum_{y \in \{0,1\}} \sqrt{m_{x,y}(V)} \log(m_{x,y}(S))$$

To obtain the finite set $\mathcal{X}$, we discretize our 2-dimensional features into 56 boxes. The square-root in the expression above does not occur in the original paper and was introduced by us due to better empirical performance. The intuition for constructing $f(\cdot)$ in this way is that the feature $x$ should appear alongside label $y$ in the chosen subset with roughly the same frequency as in the ground training set.