# OpenReview forum: "Efficient Interactive Maximization of BP and Weakly Submodular Objectives"
_auai.org/UAI/2024/Conference — UAI 2024 poster_

### Official Review · Reviewer_Zn22 · 2024-03-14

**Q2-1 Originality-Novelty:** 2
**Q2-2 Correctness-Technical Quality:** 2
**Q2-5 Clarity Of Writing:** 2

**Q1 Summary And Contributions:**

Please see below. (It didn't fit here - the character limit was exceeded).

**Q2-3 Extent To Which Claims Are Supported By Evidence:**

2: Fair: the main claims are somewhat supported by evidence (but the experimental evaluation may be weak, or does not match entirely with the claims, important baselines may be missing, proofs contain important ideas but lack rigor, algorithmic details are only discussed superficially, references are imprecise, assumptions are not sufficiently motivated or explicated, etc.).

**Q2-4 Reproducibility:**

2: Fair: key resources (e.g. proofs, code, data) are unavailable but key details (e.g. proof sketches, experimental setup) are sufficiently well-described for an expert to confidently reproduce the main results.

**Q3 Main Strengths:**

SUMMARY AND CONTRIBUTIONS

The submission builds on several works. The closest one is perhaps [Chen et al. 2017] which considers the following problem. There are m users, each user i has a monotone submodular function h_i over ground set V. At time t the environment selects user i=u_t, the optimizer then chooses an item v_t \in V to be added to the i'th user, and finally the environment reveals the marginal gain of function h_i for such addition (plus noise). Crucially, functions h_i are not known to the optimizer; instead, the optimizer knows only feature vector \phi_i characterizing h_i, and functions h_i(\phi_i, ...) are assumed to live in a Reproducing Kernel Hilbert Space and to have a bounded norm. [Chen et al.] presented an algorithm that achieves a small regret; it essentially combines the Upper-Confidence-Bound (UCB) bandit algorithm with a greedy algorithm for submodular function maximization.

The submission uses exactly the same framework, but considers more general functions h_i: (a) "BP functions" (monotone submodular plus monotone supermodular), and (b) "weakly submodular functions". Offline greedy algorithms for such maximizing such functions have been analyzed in [Bai-Bilmes 2018] and in [Bian et al. 2019], respectively. The authors extend the analysis in these papers to the case when functions are known only up to an additive error. Using this, they then further extend the results to the setting of [Chen et al. 2017] and obtain small regret bounds.

The submission also considers the case when h_i=f_i+g_i is a BP function, but the environment reveals marginal gains for f_i and g_i separately. They provide an alternative algorithm that greedy maximizes a "distorted objective" used in [Liu et al. 2022a] instead of the original objective. The authors claim that the analysis in [Liu et al. 2022a] has a mistake, and provide a corrected approximation factor for the offline case and for their bandit scenario.

Finally, the authors address efficiency of the UCB algorithm by adopting a Nystroem method from [Zenati et al. 2022], where it was used in a similar scenario but without submodularity.


MAIN STRENGTHS

The authors combined quite technical results from [Chen et al. 2017, Bai-Bilmes 2018, Bian et al. 2019, Liu et al. 2022a, Zenati et al. 2022], and additionally found and corrected a mistake in [Liu et al. 2022a]. On a technical level, this looks like an impressive achievement.

**Q4 Main Weakness:**

I found a few mistakes and inaccuracies (see below). Complete verification of all claims does not seem feasible; several of the proofs (in the appendices) refer rather vaguely to previous papers, or use phrases like "we can follow the same argument ...".
The experiments are conceptual, and some important details are not clearly described. The authors argue that for some tasks it is important to have the BP objective (submodular + supermodular); however their Algorithm 4 for this case assumes that there is a separate feedback for each term, and it was not clear to me whether proposed applications (Vignettes 1 and 2) allow that.

**Q5 Detailed Comments To The Authors:**

- Mistake in Liu et al. [2022a]: did you confirm this with the authors? If not, then more details may be needed for the claim in Appendix F.5 that the inequality in Liu et al. "does not hold in general" (e.g. a counterexample).

- Two lines before Lemma 3: \pi => \pi_j (twice)

- Two lines before Sec. 4.1: k((v,S,\phi),...) => k((\phi,S,v),...)

- Equation after eq. (7): \tilde k_{G'}(x_t) = M^T k_{G'}(x_t)
The dimensions don't seem to match. M is a matrix of size t times t, while k_{G'}(x_t) is a vector of size |G'| << t.
In general, this description was not easy to read. The definitions of \hat_t and M seem to be mutually recursive;
to make this more explicit, you can use M_t instead of M.

- The notation in Algorithm 1 is not very consistent. In particular, y_t is a vector but X_t is a set, whereas elements of y_t and X_{t-1} should match to each other.
Why not define X_{t-1}=[x_1,...,x_{t-1}] in line 8, as done for y_t?

I was also confused by the use of lists in Algorithms 2 and 3. It seems these lists are not used explicitly in the equations, but rather indicate which quantities should be stored. One could just write a remark about this instead of using these lists.

- Assumption (2a) does not seem realistic; I cannot imagine a scenario in which one would know the modular lower bound but not the function itself. Therefore, I suggest that the authors focus on the case without this assumption, and incorporate the modification in Section F.7 directly in Algorithm 4.
- Assumption (2b) does not seem to be compatible with Vignettes 1 and 2; can you comment?
- Assumption (2c): as far as I understood from Appendix F.8, there are no formal guarantees on the regret withouth this assumption. Section 5.1 should make it more clear.



Experiments:
- I did not understand what exactly the authors are doing in their experiments. Are you using hand-crafted submodular + supermodular objectives, and pretend that they are unknown to the optimizer? The authors provide Alg. 1 and several versions of Alg. 4; it was not clear to me which ones were used.

For each experiment please specify the precise form of each function in the BP objectives, whether they are assumed to be known to be the optimizer, and which algorithms are being compared. It would be best not to use Assumption (2a) (and 2c, if possible). If the objectives are hand-crafted but assumed to be unknown to the optimizer, it would be good to discuss which claims such experiments support.



Appendix E:

- The proof of Lemma 1 appeals to Lemma 4, but Lemma 4 is given only later. In general, it was not easy to navigate the numeration of lemmas / theorem / definitions due to overlap of numbers (and in particular between the main part and the appendix).

- Proof of Lemma 1:
-- definition of \omega after eq. (14): did you forget factor 1/\kappa_f? Otherwise it does not match the claim of Lemma 1.
-- "... they follow direction from the arguments in Bai and Bilmes [2018]". Is it possible to be more precise, e.g. cite a specific theorem?
-- Two lines later: T(C) \le ... => T(C,\kappa ...) \le

- Quantity a_j in Lemma 4 is not defined.

Appendix F.5: "Remarks on Liu et al. [2022b]" => "Remarks on Liu et al. [2022a]"

**Q9 Complying With Reviewing Instructions:**

Yes

---

> ### Author Rebuttal · Authors · 2024-04-06
>
> ## Rebuttal for Reviewer Zn22, Part 1 of 2
>
> Thank you for the detailed feedback and questions - it is much appreciated. Please find responses to each of your queries below.
>
> **Mistake in (Liu et al. 2022a)**
>
> > More details may be needed for the claim in Appendix F.5 that the inequality in Liu et al. "does not hold in general" (e.g. a counterexample).
>
> In (Liu et al., 2022a) on the bottom of Pg.188, they use the inequality
>
> $$ \sum_{e \in \text{OPT}} g_1 (e | S_t) \geq (1 - \kappa^g) (g_1(OPT) - g_1(S_t)) $$
>
> This inequality would follow from Lemma C.1 (iv) from (Bai and Bilmes, 2018) if the quantity $\kappa^{g}$ was swapped with $\kappa^{g_1}$; this is probably the reason the error occurred in the first place.  Consider the following counterexample with $|V| = 3$ and $k = 2$ as the cardinality constraint.
>
> Define $g(S) = |S|^2$, which is a concave over modular function, so it is supermodular. We can verify from definitions that $\kappa^g = 0.8$ and the modular lower bound $l_2(S) = |S|$, so that $g_1(S) = |S|^2 - |S|$.
>
> For simplicity, consider the case where $t=0$, so that $S_t = \emptyset$. Then, plugging into the equation, we see that the LHS is $0$, whereas the RHS is $0.2 \times 4 = 0.8 > 0$. Hence, this is a contradiction.
>
> This example can be easily generalized to any concave over modular function, larger ground set sizes or different $t$. We will include this counterexample in the final version of the paper.
>
>
> **Algorithm Description**
>
> >Two lines before Lemma 3: \pi => \pi_j (twice).
> >Two lines before Sec. 4.1: k((v,S,\phi),...) => k((\phi,S,v),...).
> >y_t is a vector but X_t is a set.
>
> Thank you for catching the typos re $\pi_j$ and the inputs to the kernel function $k(\cdot)$. We agree that it makes more sense for consistency to define $X_t$ as a vector of tuples instead.
>
>
> >$\tilde k_{G'}(x_t) = M^T k_{G'}(x_t)$ The dimensions don't seem to match. $M$ is a matrix of size $t \times t$, while $k_{G'}(x_t)$ is a vector of size $|G'| << t$.
>
> Thank you for catching this. There is a slight inaccuracy in the definition of $M$, which should only contain entries corresponding to each of the Nystrom anchor points and not all points. We will use the notation $M_t$ as you suggest. We will be sure to make these fixes in the final version and expand the explanation of the Nystrom sampling to make it more readable.
>
> >It seems these lists are not used explicitly in the equations, but rather indicate which quantities should be stored.
>
> Note that Equation (8) uses the older value $\Lambda_{t-1}$ (from L2), Equation (10) uses $K_{G_{t-1} G-{t-1}}$ (from L1) and line (9) of Algorithm 3 uses $\mathbf{\tilde{y}}_{t-1}$ (from L3); this is why we store these quantities in the lists for future use. We decided to make this very explicit in the algorithm in the interest of accuracy. There is generally a tradeoff between accuracy and simplicity, and for the pseudocode we leaned on the side of providing all the details so the algorithm can be easily re-implemented. Given the additional one page of space for the camera ready, we’ll say in the paper precisely where each list is used where (as we have done in this paragraph) should the paper be accepted.
>
> **Assumptions for Separate Feedback result**
>
> >Assumption (2a) does not seem realistic; I cannot imagine a scenario in which one would know the modular lower bound but not the function itself.
>
> For Assumption 2(a), the modular lower bound is the summation over items for the minimum possible submodular gain of selecting that item. For the case of movie recommendation, there is likely to be some marginal enjoyment from watching a movie, even if all other movies in the set have already been watched. Similarly, for the case of training set selection, for any given point, there may be some improvement in performance of the trained model due to this point even if all other points have already been included; this of course would depend on the overall size of the ground set.  We could imagine that domain knowledge could indicate what these “least gain” quantities could look like - if we are unsure, we can always choose a conservative estimate and the bound would degrade smoothly. Note also that the modular lower bound is defined by $|V|$ function evaluations, whereas the submodular function is defined by $2^|V|$. Overall this will be very application dependent.
>
> In many practical situations where the ground set is large, this modular lower bound may be close to 0 and the guarantee reduces to the one from Section F.7. That is the nature of all curvature-based bounds. In general, our philosophy with presenting the results was to provide a comprehensive set of results for a range of assumptions - then the practitioner can decide which assumptions their application is likely to satisfy and consume the corresponding result.

---

### Official Review · Reviewer_23Qa · 2024-03-21

**Q2-1 Originality-Novelty:** 2
**Q2-2 Correctness-Technical Quality:** 3
**Q2-5 Clarity Of Writing:** 3

**Q10 Ethical Concerns:**

No.

**Q1 Summary And Contributions:**

This paper considers the problem of (1) interactive online BP maximization, where the overall utility can be additively decomposed to a monotone submodular and a monotone supermodular term; (2) online weakly submodular maximization. In the BP case, the paper study the first the "monolithic feedback model" (only the overall objective can be seen) and then extend to "separate feedback" model (submodular and supermodular terms can be observed separately). For both problems, the paper provides sublinear regret guarantees.

**Q2-3 Extent To Which Claims Are Supported By Evidence:**

3: Good: the main claims are supported by convincing evidence (in the form of adequate experimental evaluation, proofs, (pseudo-)code, references, assumptions).

**Q2-4 Reproducibility:**

3: Good: key resources (e.g. proofs, code, data) are available and key details (e.g. proofs, experimental setup) are sufficiently well-described for competent researchers to confidently reproduce the main results.

**Q3 Main Strengths:**

1. The paper is well written and the problem is well motivated.
2. The paper rigorously presents both theoretical and numerical findings, as well as thorough discussions that delve into their significance and implications.

**Q4 Main Weakness:**

See Q5.

**Q5 Detailed Comments To The Authors:**

1. **Novelty.** It appears that the paper heavily builds upon the work of (Chen et al. 2017). If I understand correctly, the core part is replacing the analysis in (Nemhauser et al.,1978) with that of (Bai and Bilmes,2018) and then fit into the framework of (Chen et al. 2017). From this aspect, it seems the contribution is incremental.
2. **Regarding regret definition.** It looks like the regret definition is often referred to as "simple regret" in MAB literature, since the term is not accumulated over $T$ rounds, but rather is the quality of the action picked in the final round for each context $q$.
3. **Assumption 2.** It looks like this assumption is rather strong in practice, thus lead to the question of whether the "separate feedback" model is indeed practical in real-world scenarios.
4. **Missing related works.** There are some other work in online submodular maximization that has not been discussed, e.g., An Online Algorithm for Maximizing Submodular Functions, Streeter and Golovin, 2018; A framework for adapting offline algorithms to solve combinatorial multi-armed bandit problems with bandit feedback, Nie et al,2023, etc. In particular, the second paper also mentions the robustness of an offline algorithm.

**Q9 Complying With Reviewing Instructions:**

Yes

---

> ### Author Rebuttal · Authors · 2024-04-06
>
> ## Rebuttal for Reviewer 23Qa, Part 1 of 2
>
> Thank you for the review and questions. Please find answers to your questions below.
>
>
> **1. Comparison to (Chen et al., 2017)**
>
> > It appears that the paper heavily builds upon the work of (Chen et al. 2017).
>
> We would like to highlight that our work extends that of (Chen et. al, 2017) in a number of crucial ways. Firstly, we extend to much broader classes of functions - namely, BP functions and weakly submodular (WS) functions, which make the results much more general. Studying the robustness in the BP or WS case is much more challenging than the purely submodular case as done in (Chen et al., 2017). We do perform a much simpler analysis that ignores the submodular curvature in Appendix D, which is the analysis of (Chen et al., 2017), but this provides a much weaker approximation ratio.
>
> Instead, the analyses in Section 3.1 follow by studying the robustness of the arguments in (Bai and Bilmes, 2018) and (Bian et al. 2019) to additive errors in the greedy selection. The proofs in these papers construct an intricately designed sequence of linear programs inspired by (Conforti & Corneujols, 1984), and we study the sensitivity of these linear programs in Appendix E. In particular, we must ensure that the additive errors do not cause the solution of the LP to explode, which is not obvious apriori.
>
> Additionally, we consider a novel separate feedback model in Section 5, which is not considered by (Chen et. al, 2017). This requires us to correct an error in (Liu et al., 2022a) and extend to the online setting. Further details are in Appendix F.8
>
> Further, our algorithm employs Nystrom sampling and is faster than that of (Chen et al., 2017), whilst achieving similar $\sqrt{T}$ regret bounds. Further details in Section 4.
>
> **2. Regret Definition**
>
> >It looks like the regret definition is often referred to as "simple regret" in MAB literature.
>
> Actually, our notion of regret is different from simple regret. In simple regret in MAB, the reward of the last selected arm is compared with that of the baseline. However, in our case, we compare the reward of the set that is incrementally constructed over all timesteps with the best set that could be constructed with full information. Hence, all selections $v_t, t \in [T]$ are reflected in the regret calculation.
>
> **3. Assumption 2**
>
> > It looks like Assumption 2 is rather strong in practice, thus lead to the question of whether the "separate feedback" model is indeed practical in real-world scenarios.
>
> For Assumption 2(a), recall that the modular lower bound is the summation over items for the minimum possible submodular gain of selecting that item: $l_{q,1}(S) = \sum_{j \in S} f_q (j|V \setminus {j})$. For the case of movie recommendation, there is likely some marginal enjoyment from watching a movie, even if all other movies in the set have already been watched. We could imagine that domain knowledge could provide for what these “least gain” quantities would look like, but they would be very application specific. We also provide an alternative (slightly weaker) theorem without this assumption in Appendix F.7.
>
> For Assumption 2(b), this could be satisfied when we have more fine-grained feedback than just a single reward. For movie recommendations, we could ask users questions like “what would your rating for this movie have likely looked like if you had not watched X?”, or (more practically) we could generalize between different but similar users some of whom had and some of whom had not watched “X”. For Assumption 2(c), we provide an informal argument in Appendix F.8, which shows why using “guess and double” techniques (which are standard) instead of knowing $T_q$ is likely to work just as well in practice.
>
> In general, we provide results for different levels of assumptions, and the practitioner has the freedom to decide which setting is most applicable for their situation. Since Theorem 2 bounds a stronger notion of regret, we see that this stronger information from Assumption 2 directly leads to improved performance.

---

### Official Review · Reviewer_qBdp · 2024-03-23

**Q2-1 Originality-Novelty:** 3
**Q2-2 Correctness-Technical Quality:** 3
**Q2-5 Clarity Of Writing:** 3

**Q1 Summary And Contributions:**

The paper studies the problem of online maximizing objective functions that are non-submodular. The problem is studied in the setting of Contextual bandits with a focus on two classe of objective functions: BP decomposible functions, that are defined as the sum of a submodular function with a supermodular one, and weak submodular functions. These classes of functions are much more expressive in representing utility functions with respect to submodular ones since they allow to model the combination of complementary and competitive relationships between objects. This more general setting allows to model situations that occur in several application domains such as recommendation systems, online marketing, personalized medicine, natural language processing.

The main contribution of the paper is an algorithm that achieves sublinear alpha-regret in the Guassian Process Contextual Bandits (GPCB) setting for BP and weakly submodular objective functions. This result is built on state-of-the-art results for the optimization of these classes of functions in the offline case. However, the extension to the online case is not trivial. The paper introduces a novel definition of feedback, called separate feedback, that may be of independent interest. Moreover, an interesting method is presented to efficiently compute the objective functions by approximating its components separately in lower-dimensional spaces. Finally, the proposed algorithm is applied and experimentally analyzed in two application cases: movie recommendation systems and training data set selection in the case of machine learning.

**Q2-3 Extent To Which Claims Are Supported By Evidence:**

3: Good: the main claims are supported by convincing evidence (in the form of adequate experimental evaluation, proofs, (pseudo-)code, references, assumptions).

**Q2-4 Reproducibility:**

3: Good: key resources (e.g. proofs, code, data) are available and key details (e.g. proofs, experimental setup) are sufficiently well-described for competent researchers to confidently reproduce the main results.

**Q3 Main Strengths:**

The paper significantly extends known optimization techniques in the setting of contextual bandits to a much larger and expressive class of objective functions. Results are interesting and known trivial and some of the technical contributions may be of independent interest.
The paper is well organized and the writing is of good quality.

**Q4 Main Weakness:**

Due to the limited space the paper is very dense and many details are moved to the appendix.

**Q5 Detailed Comments To The Authors:**

No comment.

**Q9 Complying With Reviewing Instructions:**

Yes

---

> ### Author Rebuttal · Authors · 2024-04-06
>
> ## Rebuttal for Reviewer qBdp, Part 1 of 1
>
> Thank you for the positive comments, and for the suggestions on the writing. We agree that the writing is a bit dense - since we are combining many frameworks and provide an extensive suite of results, we veered on the side of accuracy instead of simplicity. We will be sure to use the extra page to reduce density for the camera-ready submission, should the paper be accepted.

---

### Official Review · Reviewer_3MCs · 2024-03-24

**Q2-1 Originality-Novelty:** 2
**Q2-2 Correctness-Technical Quality:** 3
**Q2-5 Clarity Of Writing:** 2

**Q1 Summary And Contributions:**

The authors of this paper study the problem of online interactive machine learning with combinatorial objectives. Two specific objective functions are examined in this paper: BP decomposition and weakly submodular functions. The proposed algorithms for these cases are proved to achieve sublinear $\alpha$-regret. Additionally, numerical experiments are conducted on the MovieLens dataset to demonstrate the advantages of the proposed algorithms.

**Q2-3 Extent To Which Claims Are Supported By Evidence:**

3: Good: the main claims are supported by convincing evidence (in the form of adequate experimental evaluation, proofs, (pseudo-)code, references, assumptions).

**Q2-4 Reproducibility:**

2: Fair: key resources (e.g. proofs, code, data) are unavailable but key details (e.g. proof sketches, experimental setup) are sufficiently well-described for an expert to confidently reproduce the main results.

**Q3 Main Strengths:**

* This paper provides solid theoretical results on online interactive optimization with several different objective functions.
* There are some practical examples presented in the paper that are very helpful for understanding both the problem setup and its motivations. Besides, Numerical experiments are conducted to demonstrate the effectiveness of the results.

**Q4 Main Weakness:**

* The writing of the paper needs to be improved. There are many definitions introduced in the main paper without adequate explanation.
For example, in the second paragraph on page 2, the authors introduce the definition of the regret $\mathcal{R}$. However, the definition contains typos, and the explanation of the set $S_{T_q,q}$ is confusing.
* The claim that adapting offline algorithms with $\alpha$-approximation guarantee to an online version is challenging lacks clarity on the technical obstacles. Most of the provided proofs seem to address submodular optimization problems with robustness, which may not be difficult as the errors are in additive format.

**Q5 Detailed Comments To The Authors:**

There are two major concerns in the proof: (1). The proof in the paper may be difficult for readers to follow. More specifically, the technical overview provided in the paper is relatively brief while there are many notations introduced in the proofs without enough explanation, which makes it challenging to understand the main ideas in the proof. (2). It is unclear what new techniques or challenges the authors have overcome in their proof. While the paper offers algorithms with solid theoretical guarantees on the problem, it would be better if more detailed explanations on technical challenges in this paper .

**Q9 Complying With Reviewing Instructions:**

Yes

---

> ### Author Rebuttal · Authors · 2024-04-06
>
> ## Rebuttal for Reviewer 3MCs, Part 1 of 2
>
> Thank you for the review and questions. Please find answers to your questions below.
>
>
> **Regret definition**
>
> >For example, in the second paragraph on page 2, the authors introduce the definition of the regret. However, the definition contains typos, and the explanation of the set $S_{T_q, q}$ is confusing
>
>
> Thank you for catching the typo on page 2: the $f_{\phi_m}$ should indeed be $f_{\phi_q}$.
>
> Our notion of regret is defined formally for each setup in Equations (5), (6), (12) respectively. Each of these compares the rewards of the final selected sets $S_q$ for each context with that of the best polynomial-time offline selection for that problem; here $S_q$ is the accumulated set of items for context $q$ over the entire game. The notation $S_{k, q}$ is the set selected for context $q$ of size $k$  and is chosen to be consistent with prior work (Chen et al., 2017). We agree that the notation $S_{T_q,q}$ is a bit cumbersome and in fact we considered changing it in an earlier draft of the manuscript, but kept it for several reasons: (1) It is consistent with prior work (Chen et al., 2017) and there is a benefit to consistent notation across different publications since readers of both papers will find reading both easier than if there isn’t a notation change; (2) this notation is precise, we have a sequence of length $T$, we have $m$ contexts, and at time $T$ there have been $T_q$ elements selected for context $q$, so $S_{T_q,q}$ is the set that is chosen for context $q$ at time $T$ which is of size $T_q$. This thus allows us to still refer to $S_{k,q}$ which is the the set of size $k$ chosen for context $q$ with $k \leq T_q$. We use the word “frequency” in the sense of how frequent an item for context $q$ has been selected. Given the extra page, we’ll reduce the density and offer this expanded explanation in the next version of the manuscript.
>
> **Robust offline guarantees in Section 3.1**
>
> >The proof in the paper may be difficult for readers to follow… makes it challenging to understand the main ideas in the proof.
>
> During offline optimization, we are able to compute and select the greedy optimizer at each stage. However in the online case, this is not possible since we do not have the true objective and only an approximation of it - hence, the best we can do is to select an “approximately” greedy item instead. Section 3.1 eliminates the possibility for BP and WS functions that these pointwise additive errors would explode in the final solution.
>
> >Most of the provided proofs seem to address submodular optimization problems with robustness, which may not be difficult as the errors are in additive format.
>
> We would like to highlight that studying the robustness in the BP or WS case is much more technically challenging than the purely submodular case as done in (Chen et al., 2017). In fact, we performed a simpler analysis that ignores the submodular curvature in Appendix D, which follows the (Chen et al., 2017) analysis, but this leads to a much weaker approximation ratio.
>
> Instead, our analyses follows by studying whether the arguments from (Bai and Bilmes, 2018) and (Bian et al. 2019) can be made robust to additive errors in the greedy selection. The proofs in these papers construct an intricately designed sequence of linear programs inspired by (Conforti & Corneujols, 1984), and we study the sensitivity of these linear programs in Appendix E (we are unaware of anyone who has done this before), and we have done this in the non-submodular case (BP and WS). In particular, we must ensure that the additive errors do not cause the solution of the LP to explode, which is not obvious apriori.
>
> >While there are many notations introduced in the proofs
>
> To make tracking the notation easier, we will include a “Table of Notation” at the beginning of the Appendix in the final version.

---

### Meta-Review · Area_Chair_cPfU · 2024-04-16

This paper considers an online interactive learning problem modeled as a contextual multi-armed bandit with non-linear rewards, specifically (i) submodular + supermodular rewards and (ii) weakly submodular rewards over sets of arms chosen for each context.  They propose an algorithm and prove sub-linear regret bounds.  The reviews were all on the positive side, with a number of the concerns related to presentation (such as regarding notation, high-level discussions of the challenges, elaborating on motivating applications to justify feedback models, and clarifying relations to prior works).  The authors clarified a number of points during the rebuttal.